# DP-2Stage: Adapting Language Models as Differentially Private Tabular Data Generators

**Tejumade Afonja**                                                    *tejumade.afonja@cispa.de*
*CISPA Helmholtz Center for Information Security*

**Hui-Po Wang**                                                          *hui.wang@cispa.de*
*CISPA Helmholtz Center for Information Security*

**Raouf Kerkouche**                                                *raouf.kerkouche@cispa.de*
*CISPA Helmholtz Center for Information Security*

**Mario Fritz**                                                             *fritz@cispa.de*
*CISPA Helmholtz Center for Information Security*

**Reviewed on OpenReview:** *https://openreview.net/forum?id=6nBIweDYzZ&noteId=nqBVYO1mmd*

## Abstract

Generating tabular data under differential privacy (DP) protection ensures theoretical privacy guarantees but poses challenges for training machine learning models, primarily due to the need to capture complex structures under noisy supervision signals. Recently, pre-trained Large Language Models (LLMs) – even those at the scale of GPT-2 – have demonstrated great potential in synthesizing tabular data. However, their applications under DP constraints remain largely unexplored. In this work, we address this gap by applying DP techniques to the generation of synthetic tabular data. Our findings shows that LLMs face difficulties in generating coherent text when fine-tuned with DP, as privacy budgets are inefficiently allocated to non-private elements like table structures. To overcome this, we propose DP-2Stage, a two-stage fine-tuning framework for differentially private tabular data generation. The first stage involves non-private fine-tuning on a pseudo dataset, followed by DP fine-tuning on a private dataset. Our empirical results show that this approach improves performance across various settings and metrics compared to directly fine-tuned LLMs in DP contexts. We release our code and setup at https://github.com/tejuafonja/DP-2Stage.

## 1 Introduction

Tabular data is one of the most prevalent data types, providing structured information in rows and columns, and has been extensively used across various applications. Due to privacy concerns, tabular data cannot be directly shared. A widely adopted approach to address this issue is to train synthetic tabular data generators under differential privacy (DP) (Dwork, 2006). Different model classes have been proposed, from marginal-based statistical models (Zhang et al., 2017; McKenna et al., 2022) to prominent Generative Adversarial Networks (GANs)-based tabular data generators (Xu et al., 2019), trained with DP-Stochastic Gradient Descent (DP-SGD) (Abadi et al., 2016). These models aim to replicate the marginal distribution of the original data while preserving the utility of the synthetic data. At the same time, they enforce a strict theoretical upper bound on privacy leakage, enabling users to generate realistic data samples without compromising privacy. Despite these advancements, existing techniques continue to grapple with major challenges, such as scalability limitations and difficulties in accurately modeling marginal distributions. These difficulties can be traced back to the intricate nature of tabular data and the challenges associated with training under noisy DP conditions.

Recently, pre-trained Large Language Models (LLMs) (Radford et al., 2019; Touvron et al., 2023) have demonstrated remarkable adaptability to tasks they have never been specifically trained for (Wei et al., 2021; Chung et al., 2022; Wang & Fritz, 2024). Acting as compact knowledge bases, LLMs present a promising opportunity for developing practical tabular data generators by leveraging pre-existing knowledge – an ability not currently feasible with traditional tabular data generators.

Recent work from Borisov et al. (2023) showcased how LLMs can effectively be used for synthetic tabular data generation, by representing each cell in the format "`<key> is <value>,`". While LLMs hold promise as generative priors for tabular data, adapting them under DP constraints introduces unique challenges. In DP-SGD, noise is added to the gradients to ensure privacy, which affects all tokens, including key tokens that may not be privacy-sensitive, such as column names or structural markers (e.g., "`<key> is ,`"). This indiscriminate application of noise can disrupt the model's ability to maintain the structural integrity and semantic clarity required for tabular data generation. As a result, both the utility and fidelity of the generated synthetic data are negatively affected, highlighting the need for more targeted noise application techniques to minimize such impacts while adhering to DP constraints.

In this work, we demonstrate that directly fine-tuning LLMs under DP constraints leads to sub-optimal performance. Our findings, shown in Figure 3, reveal that LLMs struggle to generate coherent and structured text when fine-tuned directly with DP. This challenge stems from inefficient privacy budget allocation to potentially non-private elements, such as the table column name "`<key>`" or non-functional tokens used to form a valid sentence for the LLM (e.g., "`is ,`").

To address this limitation, we propose **DP-2Stage**, a two-stage fine-tuning framework for tabular data generation. In the first stage, DP-2Stage fine-tunes the pre-trained LLM non-privately on a pseudo dataset, allowing the model to learn task-specific structures and patterns without consuming the privacy budget. In the second stage, fine-tuning proceeds on the private dataset with DP constraints (see Figure 1 for an overview of the approach). By learning the structural patterns in the first stage, the DP-constrained fine-tuning can focus on preserving the privacy of the data values.

For constructing pseudo datasets in the first stage, we investigated two strategies: (1) drawing data independently from a uniform distribution using statistics constructed from the private dataset (DP-2Stage-U) and (2) using an out-of-distribution public dataset that is unrelated to the private data (DP-2Stage-O). These approaches offer varying levels of privacy protection, with DP-2Stage-O providing stronger protection as it does not require prior knowledge of the private data.

We evaluated DP-2Stage on three datasets and observed improvements in utility metric (F1-score) by 12-25%, and marginal distribution metric (Hist) by 1-3% compared to direct DP fine-tuning (DP-Standard) demonstrating better utility and fidelity. Notably, DP-2Stage-U achieved up to 21× faster inference compared to DP-2Stage-O and DP-Standard. This speedup stems from its higher similarity to private data—both in key structure and value distribution—allowing the second-stage DP fine-tuning to focus less on learning structural patterns. Additionally, we observe that fine-tuning with DP while avoiding column shuffling, as used in Borisov et al. (2023), yielded better performance under DP constraints but underperformed in Non-DP scenarios.

We summarize our contributions as follows:

- Proposed DP-2Stage, a two-stage fine-tuning framework for LLM-based tabular data generation under DP, which fine-tunes on pseudo datasets non-privately to learn task-specific structures, enabling more efficient use of the privacy budget during DP fine-tuning.

- Achieved 3–7% relative reductions in perplexity, 12–25% improvements in the utility metric (F1-score), and 1–3% relative improvements in the marginal distribution metric (Hist) compared to direct DP fine-tuning (DP-Standard). Each experiment was repeated five times per model, with four synthetic datasets generated per model to ensure robust evaluation.

- Despite the effectiveness of our approach, our findings highlight the complexity of fine-tuning LLMs under DP for tabular data generation, emphasizing the need for further research to advance privacy-preserving

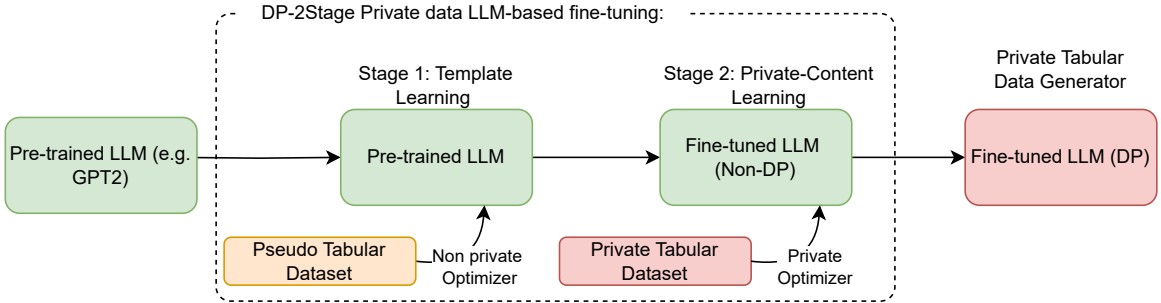

Figure 1: **Overview of DP-2Stage.** In stage 1, the pre-trained LLM is fine-tuned on the respective pseudo data. Subsequently, in stage 2, the model from stage 1 undergoes further fine-tuning using the real private data.

LLM-based tabular data generators. To support future advancements, we release our code and provide a detailed discussion to guide further investigation.

## 2 Related Work

**Tabular data generation.** As a prominent solution for data sharing, a large and growing body of research has been focused on tabular data generation. These methods can be generally categorized into marginal-based and deep learning-based methods. Marginal-based methods, such as those developed by Zhang et al. (2017); Aviñó et al. (2018); McKenna et al. (2022), view each table column as a distinct random variable and model them using specific distributions. This approach often requires prior understanding or domain expertise in the data, hindering their scalability. Nevertheless, marginal-based methods (e.g. McKenna et al. (2022)) have proven effective in integrating DP guarantees, making them a compelling choice for privacy-preserving data generation. On the other hand, deep learning (DL)-based methods such as (Choi et al., 2017; Park et al., 2018; Xu et al., 2019; Kotelnikov et al., 2023) leverage cutting-edge generative models such as Variational Autoencoders (VAEs) (Kingma & Welling, 2014), Generative Adversarial Networks (GANs) (Goodfellow et al., 2014), or diffusion models (Sohl-Dickstein et al., 2015) to synthesize tabular data. These models can capture complex patterns and correlations present in tabular data, often producing high-fidelity outputs. However, by default, these prominent DL-based tabular data generators (e.g Xu et al. (2019); Kotelnikov et al. (2023)) do not adhere to DP standards, which is a significant limitation for applications requiring strong privacy guarantees. Recent advances have introduced privacy-preserving mechanisms, such as Differentially Private Stochastic Gradient Descent (DPSGD) (Abadi et al., 2016), to modify these models for DP compliance (Xie et al., 2018). Despite this progress, ensuring DP within tabular-based deep learning models poses unique challenges due to the complexity of the preprocessing steps. For example, proposed improvements like mode-specific normalization introduced in (Xu et al., 2019) must also satisfy DP, as preprocessing steps themselves can impact privacy (Ponomareva et al., 2023). In this work, we leverage pre-trained LLMs for tabular data generation, taking advantage of their natural language processing capabilities, and investigate their potential for generating tabular data that is compliant with DP standards.

**Large Language Models (LLMs).** Language models have been extensively studied over the years, evolving from statistical models (Jelinek, 1998) and recurrent neural networks (Hochreiter & Schmidhuber, 1997) to the latest transformer-based architectures (Vaswani et al., 2017). These advancements, supported by the attention mechanisms and rich text datasets, have led to the emergence of large-scale language models (Radford et al., 2019; Brown et al., 2020; Touvron et al., 2023; Achiam et al., 2023), a new generation of language models. These models, together with various fine-tuning strategies (Hu et al., 2021; Zhou et al., 2022), have enabled several interesting applications (Borisov et al., 2023; Wang & Fritz, 2024). Borisov et al. (2023) proposed using pre-trained LLMs for synthesizing tabular data in a non-private setting by converting tabular data into text-like formats, significantly improving the performance and paving a new path toward LLM-based tabular generators. In light of the potential, our work apply these advanced models to a relatively

unexplored domain: the generation of tabular data under differential privacy constraints. While previous studies (Yu et al., 2021; Li et al., 2022) have explored private fine-tuning of LLM, our focus specifically targets tabular data generation, a unique challenge that diverges from the broader scope of text comprehension. Recently, a concurrent work (Tran & Xiong, 2024) shared a similar spirit of employing two-stage training technique to DP fine-tune LLM for tabular data generation. However, this work does not provide insight into the impact of pseudo data used in the first stage and examines a different family of LLM and fine-tuning strategy compared to this work. Our focus is on GPT-2, as utilized in the pioneering work by Borisov et al. (2023), to gain deeper insights into how effective design choices in non-differentially private (Non-DP) settings translate to DP contexts. We offer a detailed analysis of pseudo-data choice, the impact of column shuffling on generation of coherent synthetic tabular data, identify the limitations of the two-stage approach, and discuss open challenges.

## 3 Background

### 3.1 Language Models for Tabular Data

Language models are designed to model the probability of text sequences. Consider a text corpus, denoted as $\mathcal{S} = \{\boldsymbol{w}_i\}_{i=1}^N$, comprising $N$ sentences. Each sentence $\boldsymbol{w} = (t_1, \ldots, t_K)$ within $\mathcal{S}$ consists of an ordered sequence of $K$ tokens. These tokens may represent either whole words or parts of words, generated through a tokenization process such as Byte-Pair Encoding (BPE) introduced by Sennrich et al. (2015). The tokenization process can be expressed as $(t_1, \ldots, t_K) = \texttt{tokenizer}(\boldsymbol{w}_i)$. The probability of a given sentence $\boldsymbol{w}$ is formulated as:

$$p(\boldsymbol{w}_i) = p(t_1, \ldots, t_K) = \prod_{k=1}^K p(t_k|t_{<k}), \tag{1}$$

where $t_{<k} = (t_1, \ldots, t_{k-1})$ represents all tokens preceding the $k$-th token, and $p(t_k|t_{<k})$ denotes the conditional probability of token $t_k$ given all prior tokens.

In the context of tabular data, data records $\boldsymbol{w} = (\mathcal{K}, \mathcal{V})$ are defined as a collection of key-value pairs, where $\mathcal{K} = \{k_q\}_{i=1}^Q$ represents the set of keys and $\mathcal{V} = \{v_m\}_{j=1}^M$ represents the corresponding set of values. To enable LLMs to process these records, we define the serialization process below, which converts the key-value pairs into a format understandable to models.

**Definition 3.1 (Serialization)** *Let $f$ represent **template**, which defines the general pattern for organizing tabular data. Using GReaT serialization (Borisov et al., 2023), the template is expressed as $f = \text{``<key> is <value>,''}$, specifying how keys ($k \in \mathcal{K}$), values ($v \in \mathcal{V}$), and non-functional tokens ($c \in \mathcal{C}$, such as "is" and ",") combine to form a record. An input dataset denoted as $\mathcal{S}$, is a realized instance of this $f$, instantiated with tokens $(k, v)$ from a record $\boldsymbol{w} = (\mathcal{K}, \mathcal{V})$. For example, with $k = \{\texttt{age}\}$ and $v = \{32\}$, $\mathcal{S}$ is instantiated as $f(k, v) = \text{``}\texttt{age is 32},\text{''}$.*

Following tokenization, tokens associated with elements in the key or value set are denoted as $t_i \in \mathcal{K}$ or $t_j \in \mathcal{V}$, with $i$ and $j$ being the position indices, respectively. Note that, due to tokenization, the number of tokens for the keys $\mathcal{K}$ and the values $\mathcal{V}$ are not necessarily equal.

With definition 3.1, any content can be applied to a specified template.

**Column Shuffling.** Column shuffling involves randomly altering the order of column-aligned values within each batch during training. For example, given columns *age*, *sex*, and *income* with values *32*, *female*, and *>50k*. A detailed illustration is provided in Figure 2. This mechanism happens at every iteration and has been shown to effectively prevent the model from relying on spurious dependency in Non-DP settings (Borisov et al., 2023). However, it complicates DP training due to gradient perturbations, often resulting in higher perplexity and different behavior compared to Non-DP models, as shown in Figure 3. Disabling shuffling fixes the order across iterations simplifying DP training.

Example table

| age | sex | income |
|-----|-----|--------|
| 32 | female | >50k |

Iteration 1: age is 32,    sex is female,    income is >50k.
Iteration 2: income is >50k,    age is 32,    sex is female.
Iteration 3: sex is female,    income is >50k,    age is 32.

Figure 2: **Illustration of column shuffling.** The order of entries is permuted in each iteration. This mechanism happens at every iteration and has been shown to effectively prevent the model from relying on spurious dependency in Non-DP settings (Borisov et al., 2023). However, we find that it complicates DP training due to gradient perturbations, often resulting in higher perplexity compared to Non-DP models, as shown in Figure 3.

## 3.2 Differential Privacy

Differential Privacy (DP) mechanisms enable the confidential disclosure of information about a dataset by perturbing a function of the input dataset. This ensures that any information capable of distinguishing a specific record from the remainder of the dataset is constrained, as outlined by Dwork et al. (2014). In this work, we consider privacy-preserving tabular data generators to ensure that any information leakage from the generated data is bounded by DP. We review the necessary definitions, the threat model, and the privacy model below.

**Definition 3.2** *(Differential Privacy (Dwork et al., 2014)) A randomized mechanism $\mathcal{M}$ with range $\mathcal{R}$ satisfies $(\varepsilon, \delta)$-differential privacy, if for any two adjacent datasets $E$ and $E'$, i.e $E' = E \cup \{x\}$ for some $x$ in the data domain (or vice versa), and for any subset of outputs $O \subseteq \mathcal{R}$, it holds that*

$$\Pr[\mathcal{M}(E) \in O] \le e^{\varepsilon} \Pr[\mathcal{M}(E') \in O] + \delta \tag{2}$$

where $\varepsilon$ is the privacy budget and $\delta$ is the probability of the mechanism failing.

Intuitively, this guarantees that an adversary, provided with the output $\mathcal{M}$, can draw almost the same conclusions (up to $\varepsilon$ with probability larger than $1 - \delta$) about any record no matter if it is included in the input of $\mathcal{M}$ or not (Dwork et al., 2014). That is, for any record owner, a privacy breach is unlikely due to their participation in the dataset.

**Definition 3.3** *(Gaussian Mechanism (Dwork et al., 2014)) Let $f : \mathbb{R}^n \to \mathbb{R}^d$ be an arbitrary function that maps $n$-dimensional input to $d$ logits with sensitivity being:*

$$S = \max_{E, E'} \|f(E) - f(E')\|_2 \tag{3}$$

*over all adjacent datasets $E$ and $E' \in \mathcal{E}$. The Gaussian Mechanism $\mathcal{M}_\sigma$, parameterized by $\sigma$, adds noise into the output i.e.,*

$$\mathcal{M}_\sigma(x) = f(x) + \mathcal{N}(0, \sigma^2 \mathbf{I}_n). \tag{4}$$

**Threat Model.**   We outline a threat model where the goal of the adversary is to infer information about individuals in the training dataset by launching diverse privacy attacks. One such attack is the Membership Inference Attack (Shokri et al., 2017; Hilprecht et al., 2019; Chen et al., 2020), which determines whether a specific data point was included in the model's training set. We imagine a scenario involving a strong adversary who possesses complete access to the model post-training, known as "white-box access". This scenario is considerably more critical than a "black-box access" setting, where the adversary is limited to interacting with and analyzing the synthetic data produced by the model, without insight into its internal workings. In addition, the adversary is computationally unbounded. The impact of the threat model is the potential exposure of sensitive individual data from the training set, thereby compromising data privacy and undermining trust in the model.

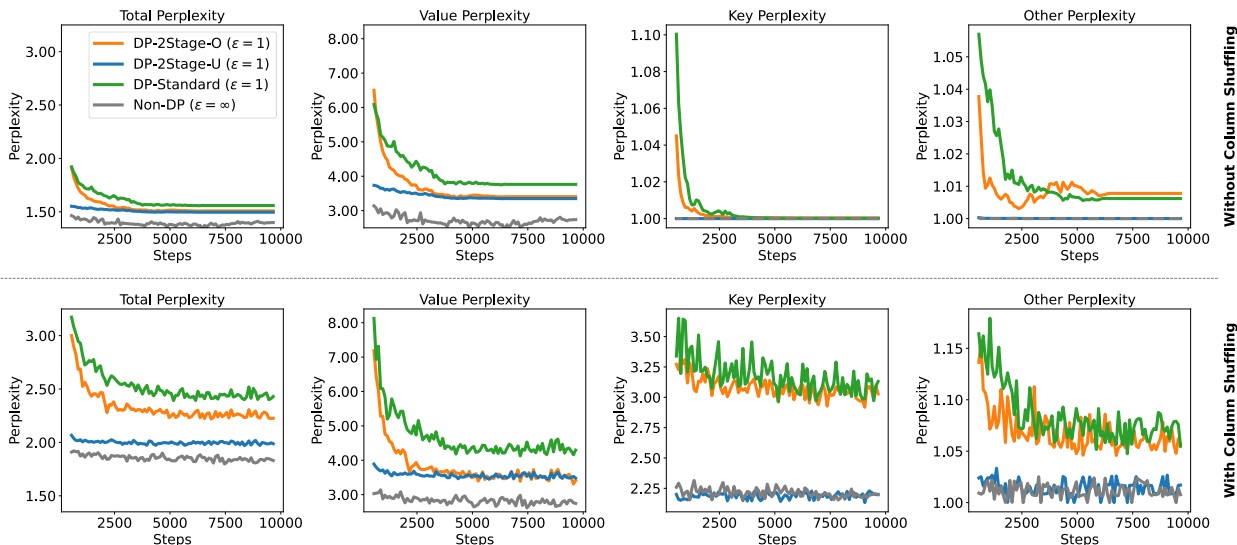

Figure 3: **DP-2Stage (Ours) vs. Standard DP fine-tuning on the Adult dataset with** $\varepsilon = 1, \delta = 10^{-5}$**.** DP-2Stage-O refers to the stage 2 model fine-tuned using out-distribution pseudo data (Airline dataset) in stage 1, while DP-2Stage-U is the stage 2 model fine-tuned using data sampled independently from a Uniform distribution, with statistics derived from the Adult dataset as the pseudo-dataset in stage 1. Perplexity results are displayed from left to right for all tokens, values, keys, and non-functional tokens (c.f. section 3.1). The top plot shows model trained without column shuffling, and the bottom shows model trained with column shuffling. Total and Value Perplexity for top and bottom plots have fixed y-axis for ease of comparison. The perplexity values are higher with column shuffling (bottom) than without (top).

**Privacy Model.** To defend against such threat, we aim to develop a solution that protects against potential privacy attacks targeting individuals in our training dataset. To achieve this, we employ DP, which was specifically designed to address this challenge. DP provides the assurance of plausible deniability, meaning that any potential privacy infringement on an individual – for example, a patient in the dataset – cannot be conclusively linked to their participation in the model's training phase up to $\varepsilon$ with probability larger than $1 - \delta$. Consider a scenario where an insurance company decides to raise a patient's insurance premium after analyzing a model trained on a medical dataset. In such a case, owing to the principles of DP, the increase cannot be directly attributed to the inclusion of that patient's data in the training process. We provide further details in Appendix A.

## 4 DP-2Stage

While large language models (LLMs) demonstrate impressive capabilities in generating synthetic tabular data in non-differentially private (Non-DP) settings, as we will later show, fine-tuning them under DP constraints remains challenging for tabular data generation. As illustrated in Figure 3 (bottom), *standard* DP training, where the LLM model is directly fine-tuned using DPSGD (Abadi et al., 2016) (see Appendix A.1 for more details), often results in suboptimal performance in terms of the perplexity metric. We hypothesize that this is due to the inefficient allocation of the privacy budget during the learning of data structures. Models employing this approach struggle to reduce the perplexity of keys (e.g., column names) and other non-functional tokens, despite these elements typically posing minimal privacy risks. Consequently, applying DP to such components not only wastes the privacy budget but also diminishes the overall utility of the model.

Motivated by this observation, we propose DP-2Stage, a two-stage learning approach designed to progressively capture tabular data structures. An overview of our method is presented in Figure 1.

In the first stage (section 4.1), we train the model on non-private *pseudo* data synthesized based on (1) prior knowledge of the private data and (2) publicly available out-of-distribution public dataset, guiding it to learn

the underlying structure. In the second stage (section 4.2), the model is fine-tuned on the private data while ensuring DP protection. The goal of our two-stage approach is to decouple the process of learning the table's structure from learning the private content.

## 4.1 Stage 1: Template Learning

The goal of the first stage is to enable the model to learn the structure of the data table, such as identifying key-value relationships, without consuming any privacy budget. We achieve it by constructing a pseudo dataset that retain the structural information while eliminating privacy risks. To achieve this, we investigated two different methods for constructing a pseudo dataset $\widetilde{\mathcal{S}}$: (1) Sampled data from a uniform distribution using the private data statistics and (2) Out-of-distribution public dataset. Each method provides different levels of reliance on private data while aiming to preserve privacy and utility.

**Independently Sampled data from a Uniform Distribution (uniform pseudo data).** This dataset is designed to *resemble* the private dataset we aim to protect. We assume that column names are public information and can be utilized without any privacy concerns. Additionally, we assume that the data owner can provide the range (maximum and minimum values) for numerical columns and the list of categories for each categorical column. For each column, samples are *independently* drawn from a uniform distribution, where all elements have an equal probability of being selected using the statistics from the private data such as the range of numerical column, and the categories in each categorical column. This assumption is minimal compared to requiring detailed distributional information, which may be unrealistic. However, while this approach minimizes reliance on private data, it still presumes access to basic information, such as category labels and numerical ranges.

**Out-of-distribution Public data (out-distribution pseudo data).** In contrast to the previous method, this approach eliminates the reliance on private data by using a publicly available dataset. In this work, we refer to *Out-of-distribution* as any publicly available dataset that is not the private data. This approach avoids the need to infer statistical properties of the original dataset, relying instead on the structure provided by the public data. As we will show, despite not requiring any information from the original dataset, this method performs comparably to uniform pseudo data in many cases and even outperforms it in certain scenarios.

**Objective function.** In the first stage, the model is trained using standard training protocols on the serialized version of the pseudo dataset. In particular, we fine-tune a pre-trained LLM $p_\theta$, parameterized by $\theta$, using cross-entropy loss for causal language modeling in a non-private setup. We formalize the loss function as follows.

$$\mathcal{L} = \mathbb{E}_{\boldsymbol{w} \in \widetilde{\mathcal{S}}} \left[ - \sum_{t_k \in \boldsymbol{w}} \log p_\theta(t_k \mid t_{<k}) \right], \tag{5}$$

where $\widetilde{\mathcal{S}}$, $\boldsymbol{w}$, and $t$ represent the pseudo dataset, individual sentences, and the corresponding tokens.

## 4.2 Stage 2: Private-Content learning

In the second stage, we fine-tune the stage 1 model to learn the private content by applying DP-SGD (Abadi et al., 2016) on the private data $\mathcal{S} = (\mathcal{K}, \mathcal{C}, \mathcal{V})$. To encourage the model to prioritize learning value tokens, we apply a weighted loss that balances the learning between value tokens and non-private tokens (key and non-functional tokens). This approach builds on the insight that the stage 1 model has already effectively learned the non-private tokens (particularly when using uniform pseudo-data) and acquired the overall structure from the first stage based on the specified template. Since Stage 1 also learns the content (key and value) while learning the template, we assign a higher weight $\lambda$ to value tokens $t \in \mathcal{V}$ in this second stage to reinforce the learning of private values as proposed by Tran & Xiong (2024). The complete loss function is defined as follows:

$$\mathcal{L} = \mathbb{E}_{(\mathcal{V},\mathcal{C},\mathcal{K})\in\mathcal{S}} \left[ -\lambda \sum_{t_j \in \mathcal{V}} \log p(t_j \mid t_{<j}) - (1-\lambda) \sum_{t_i \in \{\mathcal{K},\mathcal{C}\}} \log p(t_i \mid t_{<i}) \right], \tag{6}$$

where $\lambda$ controls the emphasis on content learning.

To summarize, the stage 2 fine-tuned LLM model is categorized based on the type of pseudo data used during stage 1. The model fine-tuned with *out-distribution* pseudo data is labeled as **DP-2Stage-O**, while the one fine-tuned with *uniform* pseudo data is labeled as **DP-2Stage-U**. In stage 1, the pre-trained LLM is initially fine-tuned on the selected pseudo data. In stage 2, this stage 1 model undergoes additional fine-tuning on the real private data (refer to Figure 1 for an overview).

## 5    Experiments

In this section, we outline the experimental protocol and evaluate our proposed method against the DP-Standard method, other Non-LLM-based DP methods, and Non-DP approaches.

### 5.1    Implementation Details

**GPT-2 Fine-tuning.**    We utilized the GPT-2 (Radford et al., 2019) model from HuggingFace[1] and integrated Opacus[2] for differential privacy (DP) training, leveraging the BatchMemoryManager for efficient micro-batching. We *fully* fine-tuned the model with DP-Adam optimizer using a learning rate of $5e{-}5$ and a linear scheduler, setting the maximum gradient norm $C = 1$ and a target delta $\delta = 10^{-5}$. In the second stage of DP-2Stage, the parameter $\lambda$ was consistently set to 0.65 across all experiments. To convert the table-to-text, we used the GReaT serialization method (Borisov et al., 2023). Unlike GReaT, we maintained a fixed column ordering for each sampled dataset for the DP. We structure the dataset so that each row begins with the target column for that dataset. The Non-DP benchmark using GPT-2, the DP-Standard model, and the stage 2 models of DP-2Stage are all trained for 10 epochs, while the stage 1 model of DP-2Stage is trained for 5 epochs.

**GPT-2 Sampling.**    To sample from the model, we condition on the target column for each dataset. For example, in the Adult dataset, where "income" is the target column, sampling begins with the prompt "income is ", and generation continues until all columns have been sampled. We use a rejection sampling approach to discard incomplete generations. However, for the ablation results in Table 4 where column shuffling is enabled, rejection sampling proved to be very slow and sometimes failed to complete any generation. In these cases, we employed an imputation-based approach, as detailed in Appendix B.1.

**Baseline Models.**    We compare our approach to Non-LLM-based methods under DP and Non-DP settings. For Non-DP baselines, we considered CTGAN, TVAE (Xu et al., 2019) and VAE (adaptation of TVAE using standardized preprocessing technique for the numerical columns i.e removing the mean and scaling to unit variance) while DP baselines include DP-CTGAN, DP-GAN, and DP-VAE. For CTGAN and TVAE, we used the implementation from the Synthetic Data Vault project[3], while DP-CTGAN and DP-GAN were implemented using the SmartNoise SDK[4]. The numerical column preprocessing was conducted with $\varepsilon = 0.1$, while the remaining privacy budget was reserved for private training. For DP-VAE, we adapted the VAE model using SmartNoise preprocessing with $\varepsilon = 0.1$.

All experiments were conducted using NVIDIA A100-SXM4-40GB GPUs. We train the models five times and generated four synthetic datasets per run. The size of the synthethic data is the same as the training data.

---

[1] https://huggingface.co/
[2] https://opacus.ai/
[3] https://github.com/sdv-dev/CTGAN
[4] https://github.com/opendp/smartnoise-sdk

### 5.1.1 Datasets

We evaluated three tabular datasets: the Adult Income dataset, which contains over 30,000 samples and 15 attributes used to predict whether an individual's annual income exceeds $50,000, hosted on the UCI Machine Learning Repository[5]; the Airline Passenger Satisfaction dataset, which includes over 100,000 samples and 24 attributes related to passenger satisfaction, available on Kaggle[6]; and the Texas Hospital Discharge dataset[7], a large publicly available dataset provided by the Texas Department of State Health Services. We used the 2013 patient dataset that is preprocessed by (Stadler et al., 2022) and available on Github[8]. Following Afonja et al. (2023), we framed the task as a binary classification problem, predicting minor vs. major mortality risk. Unlike the more commonly benchmarked Adult dataset, the Airline and Texas dataset offers a fresh perspective on the ability of deep learning models to capture complex relationships within diverse tabular data and large sample sizes. Table 1 summarizes the statistics of the tabular data. Class Ratio represents the proportion of target class categories within each dataset. We provide more details in Appendix D.1.

|         | # Train | # Test | # N | # C | # Total | Class Ratio |
|---------|---------|--------|-----|-----|---------|-------------|
| Adult   | 30932   | 16858  | 6   | 9   | 15      | 76:24       |
| Airline | 103904  | 24976  | 19  | 5   | 24      | 57:43       |
| Texas   | 60127   | 13978  | 7   | 11  | 18      | 81:19       |

Table 1: **Tabular Dataset statistics.** # N and # C are the numbers of numerical and categorical columns, respectively.

**Out-of-Distribution Public Data.** When training DP-2Stage-O on the Adult dataset, which is treated as the private dataset to be protected, we use the publicly available Airline and Texas datasets as out-of-distribution pseudo data for training the Stage 1 model. Similarly, for the Airline dataset, we use Adult and Texas, and for Texas, we use Adult and Airline as pseudo data.

### 5.1.2 Evaluation Metrics

We assess the quality of synthetic tabular datasets using a held-out test set and four key metrics inspired by prior work (Xu et al., 2019; Afonja et al., 2023; Tao et al., 2021). These metrics are categorized into two groups: Utility and Fidelity.

Utility metrics evaluate how well synthetic data preserves patterns relevant to downstream machine learning tasks. Specifically, we use (i) Machine Learning Efficacy (ML Efficacy), which compares the predictive performance of models trained on synthetic data to those trained on real data.

Fidelity metrics assess the statistical similarity between real and synthetic data. (ii) The Normalized Histogram Intersection (**HIST**) measures how closely the marginal distributions of synthetic data align with those of real data. (iii) Pairwise Correlation Similarity Accuracy (**CorAcc**) evaluates how well synthetic data preserves pairwise column correlations from the real dataset. (iv) Pairwise Attribute Distribution Similarity (**Pair**) computes the similarity of all two-way attribute distributions by averaging histogram intersections, with numerical attributes discretized into bins.

By employing these metrics, we ensure that synthetic datasets not only support effective machine learning tasks but also maintain key statistical properties of the private data.

For ML Efficacy metric, we train logistic regression[9], and XGB model[10], reporting the F1-score (**F1**), area under the receiver operating characteristic curve (**AUC**), and the accuracy score (**ACC**). For HIST and Pair, we compute averages using bins of 20 and 50. CorAcc is computed using Cramer's V (with bias correction) for

---

[5] https://archive.ics.uci.edu/dataset/2/adult
[6] https://www.kaggle.com/datasets/teejmahal20/airline-passenger-satisfaction
[7] https://www.dshs.texas.gov/thcic/
[8] https://github.com/spring-epfl/synthetic_data_release/blob/master/data/texas.csv
[9] https://scikit-learn.org/1.5/modules/generated/sklearn.linear_model.LogisticRegression.html
[10] https://xgboost.readthedocs.io/

categorical columns, Correlation Ratio for numerical-categorical columns and Pearson Correlation Coefficient (absolute values) for the numerical columns.

Additionally, for language model-based methods, we evaluate the perplexity, specifically focusing on value perplexity (**Value Perp**), which measures the model's ability to generate accurate next token while masking out the perplexity of the value tokens. More details about the metrics can be found in Appendix C.

## 5.2 Non-DP Benchmarks

Table 2 highlights the effectiveness of LLMs using GPT-2 in comparison to other benchmark methods. GPT-2, fine-tuned over 10 epochs outperforms other models in terms of utility on the Adult dataset and achieves competitive results for Pair and Hist metric. For the Airline and Texas dataset, the LLM either outperforms the GAN and VAE-based methods or achieves the second best. These results clearly demonstrate the strong performance and capability of LLMs in this context. The Real data baseline represents the optimal performance achievable, calculated by evaluating the metrics using the real training data and comparing them against the real test data.

| Dataset | Method | F1 | AUC | ACC | CorAcc | Pair | HIST |
|---------|--------|----|-----|-----|--------|------|------|
| **Adult** | Real data | $69.9_{\pm2}$ | $91.7_{\pm1}$ | $84.0_{\pm4}$ | $97.3$ | $97.5_{\pm0}$ | $99.1_{\pm0}$ |
| $\varepsilon = \infty$ | CTGAN | $59.5_{\pm6}$ | $\underline{88.5}_{\pm0}$ | $80.2_{\pm3}$ | $74.9_{\pm4}$ | $\mathbf{85.0}_{\pm1}$ | $\underline{91.2}_{\pm1}$ |
| | TVAE | $\underline{63.2}_{\pm2}$ | $87.5_{\pm1}$ | $77.7_{\pm4}$ | $\underline{76.6}_{\pm2}$ | $\underline{84.5}_{\pm1}$ | $\mathbf{91.5}_{\pm1}$ |
| | VAE | $53.8_{\pm10}$ | $86.6_{\pm1}$ | $\underline{80.5}_{\pm1}$ | $65.3_{\pm2}$ | $60.2_{\pm2}$ | $73.3_{\pm3}$ |
| | GPT-2 | $\mathbf{68.9}_{\pm0}$ | $\mathbf{90.7}_{\pm0}$ | $\mathbf{83.7}_{\pm2}$ | $\mathbf{79.9}_{\pm1}$ | $83.7_{\pm0}$ | $90.7_{\pm1}$ |
| **Airline** | Real data | $90.6_{\pm8}$ | $96.2_{\pm5}$ | $91.8_{\pm7}$ | $97.2$ | $98.4_{\pm0}$ | $99.4_{\pm0}$ |
| $\varepsilon = \infty$ | CTGAN | $\underline{87.2}_{\pm3}$ | $\underline{94.7}_{\pm2}$ | $88.9_{\pm3}$ | $\underline{84.4}_{\pm1}$ | $\mathbf{89.3}_{\pm1}$ | $\mathbf{94.4}_{\pm1}$ |
| | TVAE | $85.8_{\pm5}$ | $93.0_{\pm6}$ | $87.2_{\pm6}$ | $78.0_{\pm3}$ | $82.8_{\pm3}$ | $90.3_{\pm2}$ |
| | VAE | $79.8_{\pm1}$ | $91.1_{\pm1}$ | $80.0_{\pm1}$ | $67.8_{\pm5}$ | $60.4_{\pm1}$ | $76.8_{\pm0}$ |
| | GPT-2 | $\mathbf{89.6}_{\pm5}$ | $\mathbf{95.9}_{\pm3}$ | $\mathbf{91.4}_{\pm4}$ | $\mathbf{86.5}_{\pm4}$ | $\underline{85.5}_{\pm2}$ | $\underline{90.8}_{\pm1}$ |
| **Texas** | Real data | $86.6_{\pm2}$ | $98.8_{\pm0}$ | $95.0_{\pm1}$ | $96.3_{\pm0}$ | $98.9_{\pm0}$ | $99.5_{\pm0}$ |
| $\varepsilon = \infty$ | CTGAN | $\underline{80.8}_{\pm4}$ | $\underline{96.8}_{\pm2}$ | $\underline{92.8}_{\pm2}$ | $\mathbf{72.8}_{\pm4}$ | $\mathbf{88.3}_{\pm1}$ | $93.1_{\pm0}$ |
| | TVAE | $76.7_{\pm7}$ | $95.3_{\pm3}$ | $90.9_{\pm3}$ | $\underline{61.2}_{\pm4}$ | $63.4_{\pm7}$ | $\underline{93.5}_{\pm1}$ |
| | VAE | $74.3_{\pm3}$ | $95.6_{\pm1}$ | $90.6_{\pm2}$ | $51.5_{\pm7}$ | $57.5_{\pm1}$ | $89.3_{\pm1}$ |
| | GPT-2 | $\mathbf{83.9}_{\pm2}$ | $\mathbf{98.6}_{\pm0}$ | $\mathbf{93.6}_{\pm1}$ | $72.8_{\pm1}$ | $\underline{82.8}_{\pm3}$ | $\mathbf{94.9}_{\pm0}$ |

Table 2: **Non-DP Benchmark.** The Real data baseline represents the optimal achievable performance, determined by evaluating metrics using real training data compared to real test data. **F1**, **AUC** and **ACC** are reported as averages of two ML Models (XGBoost and Logistic Regression). **Pair**, and **Hist** are reported as averages of two bin sizes (Bins 20 and 50). Each model run five times and four synthetic datasets generated per run with standard deviation reported after $\pm$. The best value per column for each $\varepsilon$ is shown in **bold** while the second best value is underlined.

## 5.3 DP Benchmarks

Table 3 presents a comparison of DP-2Stage against GAN and VAE-based methods under strict privacy setting of $\varepsilon = 1$. Our proposed methods, DP-2Stage-U and DP-2Stage-O, demonstrate competitive performance across all metrics on all datasets. Notably, DP-CTGAN and DP-GAN show greater utility metric improvements on the Adult dataset but underperform on fidelity metrics (CorAcc, Pair, and Hist). These metrics are crucial for assessing the model's ability to preserve statistical relationships in real data, including marginal distributions and pairwise correlations.

In Appendix F.1, Table 7, we present results for a higher privacy budget ($\varepsilon = 8$). As the privacy budget increases – meaning privacy constraints are relaxed – we observe corresponding improvements in the utility metrics for DP-2Stage.

| Dataset | Method | F1 | AUC | ACC | CorAcc | Pair | HIST |
|---------|--------|-----|-----|-----|--------|------|------|
| **Adult** | *Non-LLM* | | | | | | |
| $\varepsilon = 1$, | DP-GAN | $\underline{33.5}_{\pm 20}$ | $\underline{67.7}_{\pm 9}$ | $64.2_{\pm 10}$ | $39.9_{\pm 3}$ | $41.2_{\pm 4}$ | $63.7_{\pm 3}$ |
| $\delta = 10^{-5}$ | DP-CTGAN | $\mathbf{42.2}_{\pm 20}$ | $\mathbf{78.0}_{\pm 7}$ | $\mathbf{75.7}_{\pm 3}$ | $51.3_{\pm 3}$ | $59.2_{\pm 2}$ | $75.7_{\pm 2}$ |
| | DP-VAE | $0.0_{\pm 0}$ | $50.0_{\pm 0}$ | $\underline{75.6}_{\pm 0}$ | $48.8_{\pm 1}$ | $40.3_{\pm 1}$ | $61.8_{\pm 2}$ |
| | | | | | | | |
| | *GPT-2* | | | | | | |
| | DP-Standard | $27.8_{\pm 15}$ | $58.5_{\pm 7}$ | $65.2_{\pm 9}$ | $55.0_{\pm 1}$ | $68.4_{\pm 1}$ | $85.7_{\pm 2}$ |
| | DP-2Stage-U | $21.2_{\pm 12}$ | $48.9_{\pm 6}$ | $61.9_{\pm 13}$ | $55.0_{\pm 1}$ | $\mathbf{76.1}_{\pm 1}$ | $86.7_{\pm 1}$ |
| | DP-2Stage-O | | | | | | |
| | +airline | $30.4_{\pm 17}$ | $61.6_{\pm 8}$ | $66.7_{\pm 8}$ | $\underline{55.4}_{\pm 1}$ | $\underline{72.3}_{\pm 1}$ | $\mathbf{88.5}_{\pm 1}$ |
| | +texas | $31.6_{\pm 13}$ | $60.5_{\pm 7}$ | $66.4_{\pm 8}$ | $\mathbf{55.6}_{\pm 1}$ | $71.3_{\pm 1}$ | $\underline{86.9}_{\pm 1}$ |
| **Airline** | *Non-LLM* | | | | | | |
| $\varepsilon = 1$, | DP-GAN | $40.2_{\pm 24}$ | $63.9_{\pm 13}$ | $59.8_{\pm 6}$ | $37.4_{\pm 9}$ | $22.2_{\pm 13}$ | $44.7_{\pm 12}$ |
| $\delta = 10^{-5}$ | DP-CTGAN | $\underline{67.1}_{\pm 8}$ | $\underline{76.7}_{\pm 8}$ | $\underline{68.0}_{\pm 6}$ | $31.7_{\pm 2}$ | $62.2_{\pm 2}$ | $78.7_{\pm 2}$ |
| | DP-VAE | $26.5_{\pm 28}$ | $57.9_{\pm 13}$ | $57.3_{\pm 6}$ | $46.6_{\pm 1}$ | $20.6_{\pm 0}$ | $41.8_{\pm 1}$ |
| | | | | | | | |
| | *GPT-2* | | | | | | |
| | DP-Standard | $60.5_{\pm 7}$ | $65.3_{\pm 9}$ | $62.4_{\pm 7}$ | $64.0_{\pm 2}$ | $77.0_{\pm 2}$ | $90.3_{\pm 3}$ |
| | DP-2Stage-U | $\mathbf{68.5}_{\pm 9}$ | $\mathbf{77.8}_{\pm 10}$ | $\mathbf{72.1}_{\pm 7}$ | $65.3_{\pm 1}$ | $\mathbf{80.8}_{\pm 1}$ | $\underline{90.7}_{\pm 1}$ |
| | DP-2Stage-O | | | | | | |
| | +adult | $55.2_{\pm 18}$ | $62.5_{\pm 19}$ | $60.0_{\pm 16}$ | $\mathbf{66.8}_{\pm 1}$ | $\underline{80.1}_{\pm 1}$ | $\mathbf{92.5}_{\pm 1}$ |
| | +texas | $52.5_{\pm 13}$ | $61.0_{\pm 13}$ | $58.4_{\pm 10}$ | $66.1_{\pm 2}$ | $78.7_{\pm 2}$ | $90.4_{\pm 1}$ |
| **Texas** | *Non-LLM* | | | | | | |
| $\varepsilon = 1$, | DP-GAN | $13.7_{\pm 16}$ | $58.3_{\pm 12}$ | $78.3_{\pm 8}$ | $36.1_{\pm 7}$ | $34.6_{\pm 5}$ | $68.3_{\pm 7}$ |
| $\delta = 10^{-5}$ | DP-CTGAN | $63.9_{\pm 11}$ | $91.4_{\pm 3}$ | $82.6_{\pm 19}$ | $43.9_{\pm 6}$ | $\underline{66.9}_{\pm 6}$ | $84.7_{\pm 3}$ |
| | DP-VAE | $0.0_{\pm 0}$ | $50.0_{\pm 0}$ | $82.5_{\pm 0}$ | $62.1_{\pm 1}$ | $43.9_{\pm 1}$ | $77.9_{\pm 1}$ |
| | | | | | | | |
| | *GPT-2* | | | | | | |
| | DP-Standard | $55.4_{\pm 10}$ | $90.2_{\pm 5}$ | $77.1_{\pm 11}$ | $\underline{70.3}_{\pm 1}$ | $60.6_{\pm 1}$ | $92.3_{\pm 1}$ |
| | DP-2Stage-U | $23.5_{\pm 14}$ | $59.8_{\pm 14}$ | $67.3_{\pm 17}$ | $68.2_{\pm 0}$ | $\mathbf{80.7}_{\pm 6}$ | $\mathbf{93.4}_{\pm 0}$ |
| | DP-2Stage-O | | | | | | |
| | +adult | $\mathbf{74.8}_{\pm 4}$ | $\mathbf{96.7}_{\pm 1}$ | $\mathbf{89.1}_{\pm 3}$ | $69.9_{\pm 2}$ | $60.5_{\pm 2}$ | $91.7_{\pm 1}$ |
| | +airline | $\underline{74.3}_{\pm 5}$ | $96.4_{\pm 0}$ | $88.8_{\pm 3}$ | $\mathbf{70.9}_{\pm 1}$ | $62.0_{\pm 2}$ | $93.2_{\pm 0}$ |

Table 3: **DP Benchmark.** Utility metrics (F1, AUC, and ACC) are presented as the averages of two ML Models (XGBoost and Logistic Regression). **Pair**, and **Hist** are reported as averages of two bin sizes (Bins 20 and 50). Results are averaged across five model runs and four synthetic datasets per run with standard deviation reported after $\pm$. The best value per column for $\varepsilon = 1$ is shown in **bold** while second best value is underlined.

## 5.4 DP-2Stage Benchmark

We compare our proposed DP-2Stage methods, which leverage two pseudo-dataset approaches (DP-2Stage-U and DP-2Stage-O) as described in Section 4.1, against DP-Standard, which directly fine-tunes the LLM under a strict privacy budget ($\varepsilon = 1$). Our methods consistently improve downstream task utility and fidelity. As shown in Table 3, DP-2Stage-O achieves the highest F1, AUC, and ACC scores on the Adult and Texas datasets but performs slightly worse on the Airline dataset compared to DP-2Stage-U and DP-Standard. We hypothesize that the choice of pseudo data may have influenced this outcome – Adult and Texas datasets may not provide sufficient structural information for effective second-stage DP fine-tuning on Airline. Further investigation is needed to understand the key properties required in pseudo data to optimize performance. Additionally, our approaches achieve the best results across all fidelity metrics i.e CorAcc, Pair, and Hist.

### 5.4.1 Impact of Column Shuffling

Next, we evaluate the impact of column shuffling. In Figure 3, we compare the perplexity of DP and Non-DP LLM models on the Adult dataset. The top section illustrates the evolution of perplexity with column shuffling enabled – the default setting in prior work by Borisov et al. (2023) – while the bottom section shows perplexity without shuffling. The results indicate that the non-shuffling configuration yields the lowest perplexity under DP.

Table 4 presents a detailed comparison of these two approaches on the Adult and Airline datasets. The findings suggest that column shuffling generally benefits the Non-DP setting, which may explain its widespread use in prior research. However, under DP, models without shuffling tend to achieve better performance, as highlighted by the bold values. This advantage may stem from the added complexity that shuffling introduces to DP training, potentially amplifying the effects of gradient perturbation[11]. That said, the Hist metric show better performance with shuffling except for DP-2Stage-O on Adult dataset.

Additionally, enabling shuffling significantly increases sampling time (approximately 42–51 hours per model run), particularly for DP-Standard and DP-2Stage-O (see Table 6 in Appendix E). As a result, to ensure a fair evaluation when shuffling is enabled, we report results from a single training run for DP-Standard and DP-2Stage-O.

| | Method | Shuffle | Adult | | | | | Airline | | | | |
|---|---|---|---|---|---|---|---|---|---|---|---|---|
| | | | Value Perp | F1 | AUC | ACC | HIST | Value Perp | F1 | AUC | ACC | HIST |
| GPT-2 $\varepsilon = \infty$ | Non-DP | ✓ | **2.398**$_{\pm 0.00}$ | 68.9$_{\pm 0}$ | **90.7**$_{\pm 0}$ | 83.7$_{\pm 2}$ | 90.7$_{\pm 1}$ | **2.865**$_{\pm 0.00}$ | 89.6$_{\pm 5}$ | 95.9$_{\pm 3}$ | 91.4$_{\pm 4}$ | 90.8$_{\pm 1}$ |
| | | ✗ | 2.482$_{\pm 0.01}$ | 69.0$_{\pm 1}$ | 90.7$_{\pm 0}$ | 83.3$_{\pm 2}$ | 90.2$_{\pm 0}$ | 2.901$_{\pm 0.01}$ | 76.5$_{\pm 24}$ | 91.3$_{\pm 7}$ | 84.0$_{\pm 10}$ | 93.5$_{\pm 3}$ |
| GPT-2 $\varepsilon = 1$, $\delta = 10^{-5}$ | DP-Standard | ✓ | 3.537 | 24.6$_{\pm 23}$ | **62.4**$_{\pm 8}$ | 63.9$_{\pm 14}$ | **87.4**$_{\pm 0}$ | 4.276 | 44.8$_{\pm 33}$ | **73.9**$_{\pm 12}$ | **66.9**$_{\pm 11}$ | **93.3**$_{\pm 1}$ |
| | | ✗ | **3.224** | **31.4**$_{\pm 17}$ | 62.1$_{\pm 8}$ | **68.4**$_{\pm 5}$ | 85.9$_{\pm 1}$ | **3.790** | **59.4**$_{\pm 6}$ | 65.8$_{\pm 4}$ | 61.8$_{\pm 2}$ | 88.3$_{\pm 1}$ |
| | DP-2Stage-U | ✓ | 2.973$_{\pm 0.01}$ | 19.7$_{\pm 13}$ | 48.0$_{\pm 8}$ | **61.9**$_{\pm 13}$ | **87.6**$_{\pm 1}$ | 3.932$_{\pm 0.08}$ | 48.8$_{\pm 11}$ | 57.5$_{\pm 7}$ | 55.2$_{\pm 5}$ | **93.4**$_{\pm 1}$ |
| | | ✗ | **2.887**$_{\pm 0.04}$ | **21.2**$_{\pm 12}$ | **48.9**$_{\pm 6}$ | **61.9**$_{\pm 13}$ | 86.7$_{\pm 1}$ | **3.633**$_{\pm 0.01}$ | **68.5**$_{\pm 9}$ | **77.8**$_{\pm 10}$ | **72.1**$_{\pm 7}$ | 90.7$_{\pm 1}$ |
| | DP-2Stage-O +adult | ✓ | - | - | - | - | - | 3.948 | 59.8$_{\pm 4}$ | 68.9$_{\pm 6}$ | 66.3$_{\pm 5}$ | **92.0**$_{\pm 1}$ |
| | | ✗ | - | - | - | - | - | **3.737** | **73.2**$_{\pm 2}$ | **82.0**$_{\pm 2}$ | **75.7**$_{\pm 2}$ | 91.7$_{\pm 1}$ |
| | +airline | ✓ | 3.270 | 22.7$_{\pm 22}$ | 55.1$_{\pm 10}$ | **68.0**$_{\pm 9}$ | 87.1$_{\pm 1}$ | - | - | - | - | - |
| | | ✗ | **3.049** | **27.8**$_{\pm 18}$ | **60.2**$_{\pm 7}$ | 65.5$_{\pm 9}$ | **88.4**$_{\pm 0}$ | - | - | - | - | - |

Table 4: **Comparison of Methods with Shuffle Enabled (✓) or Disabled (✗).** The best result under DP is highlighted in **bold**, while the top-performing result for each shuffle setting is underlined. Non-DP methods generally perform comparably across both settings but tend to show better results with shuffle enabled (✓). Conversely, DP methods often achieve higher performance when shuffle is disabled (✗). Results are averaged across five model runs, each using a different random seed, with four synthetic datasets generated per run. **For DP-Standard and DP-2Stage-O, results are based on a single model run**, averaging across four synthetic datasets, due to the high inference time required with shuffle enabled (approximately 42–51 hours per run; see Appendix E, Table 6), ensuring a fair evaluation.

### 5.4.2 Impact of Weighted Loss

To assess the impact of loss weighting in the second stage (see Equation 6), we conducted experiments using both the default setting ($\lambda = 0.33$) and the proposed weighting value ($\lambda = 0.65$). As shown in Table 5, emphasizing value tokens during second-stage DP fine-tuning leads to the best overall performance. This approach also improves DP-Standard, though the gains are more pronounced for DP-2Stage-O.

On the Adult dataset, DP-2Stage-U generally performs better with the default loss, except for the F1 score, whereas on the Airline dataset, it performs worse with the default loss. These results suggest that loss weighting benefits both DP-2Stage-O and DP-Standard, though its effects on DP-2Stage-U remain inconsistent, warranting further investigation.

---

[11]When column shuffling is applied during DP training, it adds complexity to the learning process. This additional complexity could make it harder for the model to learn meaningful patterns, especially under DP constraints, where noise is deliberately added to gradients to ensure privacy. Because DP already introduces randomness into the training process, shuffling may further disrupt the model's ability to learn effectively by increasing variability in the data order. This, in turn, could amplify the impact of the noise added by DP, making training less stable and reducing performance.

| | Method | $\lambda$ | Adult | | | | Airline | | | |
|---|---|---|---|---|---|---|---|---|---|---|
| | | | **F1** | **AUC** | **ACC** | **HIST** | **F1** | **AUC** | **ACC** | **HIST** |
| GPT-2 $\varepsilon = 1$ $\delta = 10^{-5}$ | DP-Standard | default | $27.8_{\pm15}$ | $58.5_{\pm7}$ | $65.2_{\pm9}$ | $85.7_{\pm2}$ | $\mathbf{60.5}_{\pm7}$ | $65.3_{\pm9}$ | $62.4_{\pm7}$ | $90.3_{\pm3}$ |
| | | 0.65 | $\mathbf{28.9}_{\pm17}$ | $\mathbf{60.9}_{\pm8}$ | $\mathbf{66.8}_{\pm8}$ | $\mathbf{87.4}_{\pm2}$ | $59.9_{\pm7}$ | $\mathbf{65.7}_{\pm9}$ | $\mathbf{62.9}_{\pm7}$ | $91.6_{\pm4}$ |
| | DP-2Stage-U | default | $20.8_{\pm14}$ | $\mathbf{49.7}_{\pm7}$ | $\mathbf{63.0}_{\pm12}$ | $\mathbf{87.3}_{\pm1}$ | $65.0_{\pm10}$ | $74.5_{\pm9}$ | $69.3_{\pm6}$ | $\mathbf{91.1}_{\pm1}$ |
| | | 0.65 | $\mathbf{21.2}_{\pm12}$ | $48.9_{\pm6}$ | $61.9_{\pm13}$ | $86.7_{\pm1}$ | $\mathbf{68.5}_{\pm9}$ | $\mathbf{77.8}_{\pm10}$ | $\mathbf{72.1}_{\pm7}$ | $90.7_{\pm1}$ |
| | DP-2Stage-O +adult | default | - | - | - | - | $48.8_{\pm17}$ | $55.6_{\pm19}$ | $54.4_{\pm15}$ | $92.4_{\pm1}$ |
| | | 0.65 | - | - | - | - | $\mathbf{55.2}_{\pm18}$ | $\mathbf{62.5}_{\pm19}$ | $\mathbf{60.0}_{\pm16}$ | $\mathbf{92.5}_{\pm1}$ |
| | +airline | default | $30.3_{\pm15}$ | $60.4_{\pm7}$ | $65.8_{\pm9}$ | $87.3_{\pm1}$ | - | - | - | - |
| | | 0.65 | $\mathbf{30.4}_{\pm17}$ | $\mathbf{61.6}_{\pm8}$ | $\mathbf{66.7}_{\pm8}$ | $\mathbf{88.5}_{\pm1}$ | - | - | - | - |

Table 5: **Comparison of $\lambda$ Values for Weighting the Stage 2 Loss.** Results are averaged over five model runs with different random seeds, and four synthetic datasets generated per run.

### 5.4.3 Understanding the Utility-Perplexity Gap in DP-2Stage-U

In Table 4, we present perplexity values with and without column shuffling alongside utility performance metrics. While uniform pseudo-data (DP-2Stage-U) often achieves lower perplexity, this does not always translate to improved downstream utility or histogram scores. In some cases, the out-distribution pseudo-data (DP-2Stage-O) approach yields better performance. We hypothesize that this discrepancy observed in DP-2Stage-U may result from overfitting when pseudo-data closely resembles the private data. In such cases, the model may lack sufficient incentive to learn meaningful correlations in the private data during the second stage. Conversely, when out-distribution pseudo-data is used (DP-2Stage-O), the model is encouraged to learn these correlations while retaining the structural information established during stage 1. This approach may provide better overall performance compared to directly applying DP. Notably, DP-2Stage-U achieves the fastest inference time – up to 21x faster (see Appendix E, Table 6) – making it particularly advantageous for real-world deployment.

### 5.4.4 Marginal-based DP-Baselines.

In Appendix F.2, we present results for two marginal-based DP methods: MST (McKenna et al., 2021) and AIM (McKenna et al., 2022). As shown in Appendix F.2, Table 8, these methods achieve strong performance across most metrics. However, they underperform on certain fidelity metrics, such as Pair and Hist, particularly for the Airline dataset – likely due to the large number of numerical columns in this dataset.

## 6 Limitations & Future Work

**Stage 1 Training Iterations.** In our proposed DP-2Stage approach, we trained the stage 1 model for 5 epoch, as done in concurrent work Tran & Xiong (2024). However, further exploration is needed to assess the impact of varying the number of training iterations and identifying an optimal checkpoint during stage one. This could enhance the performance of stage two by providing a more refined initialization point. Future research could also investigate (1) alternative strategies for constructing the stage one dataset, and (2) how variations in stage one datasets influence DP fine-tuning performance in stage two.

**Serialization Methods.** The serialization method employed in this work, based on the approach proposed by Borisov et al. (2023), has not been compared with alternative methods. Exploring different serialization strategies could reveal approaches that are better suited for DP, potentially leading to enhanced performance. Future research could investigate the impact of various serialization methods to identify those that most effectively address the needs of DP.

**DP Hyperparameters Tuning.** This work did not involve an extensive exploration of hyperparameters, such as batch size, learning rate schedulers, learning rate, or clipping strategies, which are critical for DP

performance. Conducting a systematic analysis of these hyperparameters could yield valuable insights into optimizing LLMs for use as DP tabular data generators. Future research in this direction could help unlock the full potential of LLMs in this domain.

**Scaling to Larger Models.** Exploring the use of larger models presents a promising avenue for enhancing performance. However, this work did not investigate this direction due to the significant computational costs associated with training and inference for models with a high number of parameters. Balancing the trade-off between performance gains and computational feasibility remains a critical challenge. Future research could focus on developing strategies to mitigate these costs, enabling a more practical evaluation of larger models.

**Privacy Guarantees (in Stage 1).** In Stage 1, assuming that the public dataset used is indeed "public", the privacy guarantees of our method hold exactly, as only Stage 2 is exposed to private data and is trained with differential privacy. This is the case for our DP-2Stage-O method.

However, in Stage 1, the proposed DP-2Stage-U method utilizes private data to compute statistics for the uniformly sampled dataset, which weakens the stated privacy guarantees. To ensure that the privacy guarantees from the second stage hold completely, these statistics should instead be computed in a differentially private manner by allocating a small privacy budget. Future work should explore this approach.

## 7 Conclusion

In this work, we investigated the use of pre-trained LLMs for tabular data generation under DP protection. Our analysis indicates that naïvely fine-tuning the model results in sub-optimal performance due to inefficient allocation of privacy budgets. In light of this, we propose DP-2Stage, a two-stage optimization approach that first adapt the model to the format of the task in the first stage and proceeds with fine-tuning the model for the DP task. This two-stage strategy secures robust privacy protection while ensuring that privacy budgets are spent on the actual sensitive data, leading to improved data utility and efficient data generation. Despite the competitive performance and increased efficiency, further investigation is needed to optimize privacy budget allocation and improve scalability, as discussed in section 6.

**Broader Impact Statement**

This research examines the privacy risks of using pre-trained LLMs, such as GPT-2, for generating tabular data and introduces a differentially private fine-tuning process to address these concerns. While this approach reduces re-identification and data leakage by incorporating DP into the training process, it also incurs significant environmental costs due to the high computational demands. For example, training a non-differentially private CTGAN for 300 epochs takes around 10 minutes, whereas fine-tuning GPT-2 for 10 epochs requires approximately 2 hours on the same hardware and dataset, with DP training extending the time further due to per-example gradient computations. This disparity underscores the greater computational expense of LLMs for both training and inference compared to traditional GAN-based tabular data generators. We hope to inspire dialogue on how to leverage the capabilities of LLMs for tabular generation task in a more sustainable and environmentally conscious manner.

**Reproducibility Statement**

To ensure reproducibility, we outline several key efforts throughout the paper. The methodology for our proposed DP-2Stage framework is detailed in section 4, and section 5 describes the public datasets and open-source model used in our experiments. Additionally, we have released the source code to enable further experimentation and validation of our findings https://github.com/tejuafonja/DP-2Stage.

**Acknowledgement**

This work is supported by ELSA – European Lighthouse on Secure and Safe AI funded by the European Union under grant agreement No. 101070617, Bundesministeriums fur Bildung und Forschung (PriSyn), grant No. 16KISAO29K, and Medizininformatik-Plattform "Privatsphärenschutzende Analytik in der Medizin" (PrivateAIM), grant No. 01ZZ2316G.

Computation resources used in this work are supported by the Helmholtz Association's Initiative and Networking Fund on the HAICORE@FZJ partition and CISPA Helmholtz Center for Information Security computing services. Views and opinions expressed are those of the authors only and do not necessarily reflect those of the European Union or European Commission. Neither the European Union nor the European Commission can be held responsible for them. We thank the TMLR reviewers and action editor for their detailed feedback, which has undoubtedly improved the quality of our work. We also thank Joscha Cüppers for providing valuable feedback on this work.

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
