# A  Differential Privacy: more details

## A.1  Standard DP

We define the *Standard* DP or **DP-Standard** training process as fine-tuning the LLM directly on private data using the Differentially Private Stochastic Gradient Descent (DPSGD) (Abadi et al., 2016) mechanism. DPSGD ensures that the training process complies with the formal definition of differential privacy (refer to Definition 3.2) through the following steps: (1) Gradients are computed for each individual sample within a mini-batch. (2) Gradients are clipped to a fixed norm to bound sensitivity. (3) Gaussian noise, calibrated to the privacy parameters $(\varepsilon, \delta)$, is added to the aggregated gradients. (4) The resulting noisy gradients are used to update the model's parameters.

## A.2  Privacy Analysis

**Definition A.1** *(Rényi divergence). Let $P$ and $Q$ be two distributions on $\mathcal{X}$ defined over the same probability space, and let $p$ and $q$ be their respective densities. The Rényi divergence of a finite order $\alpha \neq 1$ between $P$ and $Q$ is defined as follows:*

$$D_\alpha(P \parallel Q) \triangleq \frac{1}{\alpha - 1} \ln \int_{\mathcal{X}} q(x) \left( \frac{p(x)}{q(x)} \right)^\alpha dx \tag{7}$$

*Rényi divergence at orders $\alpha = 1, \infty$ are defined by continuity.*

**Definition A.2** *(Rényi differential privacy (RDP)). A randomized mechanism $\mathcal{M} : \mathcal{E} \to \mathcal{R}$ satisfies $(\alpha, \rho)$-Rényi differential privacy (RDP) if for any two adjacent inputs $E, E' \in \mathcal{E}$ it holds that*

$$D_\alpha(\mathcal{M}(E) \parallel \mathcal{M}(E')) \leq \rho \tag{8}$$

*In this work, we call two datasets $E, E'$ to be adjacent if $E' = E \cup \{x\}$ (or vice versa).*

**Definition A.3** *(Sampled Gaussian Mechanism (SGM)). Let $f$ be an arbitrary function mapping subsets of $\mathcal{E}$ to $\mathbb{R}^d$. We define Sampled Gaussian mechanism (SGM) parameterized with the sampling rate $0 < q \leq 1$ and the noise $\sigma > 0$ as*

$$SG_{q,\sigma} \triangleq f(\{x : x \in R \text{ is sampled with probability } q\}) + \mathcal{N}(0, \sigma^2 \mathbf{I}^d) \tag{9}$$

*where each element of $E$ is independently and randomly sampled with probability $q$ without replacement. The sampled Gaussian mechanism consists of adding independent and identically distributed (i.i.d) Gaussian noise with zero mean and variance $\sigma^2$ to each coordinate value of the true output of $f$. In fact, the sampled Gaussian mechanism draws vector values from a multivariate spherical (or isotropic) Gaussian distribution which is described by random variable $\mathcal{N}(0, \sigma^2 \mathbf{I}^d)$, where $d$ is omitted if it unambiguous in the given context.*

### A.2.1  Analysis

The privacy analysis of our DP methods and other DP baselines considered in the paper follows the well-established analysis framework used for gradient-based, record-level DP methods, known as DP-Stochastic Gradient Descent (DP-SGD) (Abadi et al., 2016). In this framework, each update is conducted as a single SGM step (Definition A.3), which includes selecting a random batch, clipping the per-example gradients of that batch, and then adding Gaussian noiseto the aggregrated batch gradient. The privacy cost accumulated over multiple updates is quantified using the revised moment accountant method (Mironov et al., 2019), which adapts the original moment accountant approach introduced by Abadi et al. (2016) to the concept of Rényi Differential Privacy (RDP) (Definition A.2). Finally, to achieve interpretable results and allow for transparent comparisons with established methods, the privacy cost is converted from $(\alpha, \rho)$-RDP to $(\varepsilon, \delta)$-DP using the conversion theorem (Theorem A.6) provided by Balle et al. (2020).

Let $\mu_0$ denote the pdf of $\mathcal{N}(0, \sigma^2)$ and let $\mu_1$ denote the pdf of $\mathcal{N}(1, \sigma^2)$. Let $\mu$ be the mixture of two Gaussians $\mu = (1 - q)\mu_0 + q\mu_1$, where $q$ is the sampling probability of a single record in a single round.

**Theorem A.4** *(Mironov et al., 2019). Let $\text{SGM}_{q,\sigma}$ be the Sampled Gaussian mechanism for some function $f$ and under the assumption $\Delta_2 f \leq 1$ for any adjacent $E, E' \in \mathcal{E}$. Then $\text{SGM}_{q,\sigma}$ satisfies $(\alpha, \rho)$-RDP if*

$$\rho \leq \frac{1}{\alpha - 1} \log \max(A_\alpha, B_\alpha) \tag{10}$$

*where $A_\alpha \triangleq \mathbb{E}_{z \sim \mu_0}[(\mu(z)/\mu_0(z))^\alpha]$ and $B_\alpha \triangleq \mathbb{E}_{z \sim \mu}[(\mu_0(z)/\mu(z))^\alpha]$*

Theorem A.4 states that applying SGM to a function of sensitivity (Definition 3.3) at most 1 (which also holds for larger values without loss of generality) satisfies $(\alpha, \rho)$-RDP if $\rho \leq \frac{1}{\alpha-1} \log(\max\{A_\alpha, B_\alpha\})$. Thus, analyzing RDP properties of SGM is equivalent to upper bounding $A_\alpha$ and $B_\alpha$. From Corollary 7. in (Mironov et al., 2019), $A_\alpha \geq B_\alpha$ for any $\alpha \geq 1$. Therefore, we can reformulate Equation 10 as

$$\rho \leq \xi_\mathcal{N}(\alpha|q) := \frac{1}{\alpha-1} \log A_\alpha \tag{11}$$

To compute $A_\alpha$, we use the numerically stable computation approach proposed in (Mironov et al., 2019) (Sec. 3.3) depending on whether $\alpha$ is expressed as an integer or a real value.

**Theorem A.5** *(Composability (Mironov, 2017). Suppose that a mechanism $\mathcal{M}$ consists of a sequence of adaptive mechanisms $\mathcal{M}_1, \ldots, \mathcal{M}_k$ where $\mathcal{M}_i : \prod_{j=1}^{i-1} \mathcal{R}_j \times \mathcal{E} \to \mathcal{R}_i$. If all the mechanisms in the sequence are $(\alpha, \rho)$-RDP, then the composition of the sequence is $(\alpha, k\rho)$-RDP.*

In particular, Theorem A.5 holds when the mechanism themselves are chosen based on the (public) output of the previous mechanisms. By Theorem A.5, it suffices to compute $\xi_\mathcal{N}(\alpha|q)$ at each step and sum them up to bound the overall RDP privacy budget of an iterative mechanism composed of single DP mechanism at each step.

**Theorem A.6** *(Conversion from RDP to DP (Balle et al., 2020)). If a mechanism $\mathcal{M}$ is $(\alpha, \rho)$-RDP then it is $((\rho + \log((\alpha-1)/\alpha) - (\log \delta + \log \alpha)/(\alpha-1), \delta)$-DP for any $0 < \delta < 1$.*

**Theorem A.7** *(Privacy of the different DP methods). For any $0 < \delta < 1$ and $\alpha \geq 1$, the different DP methods are $(\varepsilon, \delta)$-DP, with*

$$\varepsilon = \min_\alpha (T \cdot \xi_\mathcal{N}(\alpha|q) + \log((\alpha-1)/\alpha) - (\log \delta + \log \alpha)/(\alpha-1)) \tag{12}$$

*Here, $\xi_\mathcal{N}(\alpha|q)$ is defined in Equation 11, $q = \frac{\mathbb{B}}{|\mathcal{D}|}$, $T$ is the total number of updates, $\mathbb{B}$ is the batch size, and $|\mathcal{D}|$ denotes the dataset size.*

The proof follows from Theorems A.4, A.5, A.6 and the fact that a record is sampled in every SGDiteration if the batch of records sampled contains the record, which has a probability of at most $\frac{\mathbb{B}}{|\mathcal{D}|}$. Therefore, a record is sampled with a probability of at most $q = \frac{\mathbb{B}}{|\mathcal{D}|}$.

# B Sampling

## B.1 Imputation-based sampling

After training the model, we generate samples by conditioning on a key-value pair, i.e., $\boldsymbol{w} \sim p_\theta(\cdot \mid \texttt{Prompt})$, where `Prompt` denotes the tokens generated from the pair, for instance, tokens of "`income is <50k`". The trained model then generates the next token based on this prompt. The sampling process continues until it encounters a stop token or a maximum token length of 100, which exceeds the number of tokens in each table row. Depending on the model's performance, they may produce incoherent outputs, such as mismatches of keys and values (e.g., generating 'age is >50' and 'relationship is Ad-serv-spouse'). To this end, we post-process the generated data and remove the values that do not match the category of the corresponding column. Once removed, we use the correct tokens to recondition the model, allowing it to fill in any missing tokens — essentially performing imputation based on the correctly generated tokens. We set a threshold of 15 for imputation, meaning if the generation quality is too poor, imputation will not proceed.

Previous method (Borisov et al., 2023) often discard incorrectly generated samples and continue generating until the model produces a correct sample in one shot, or they exit the loop. While this approach works well with Non-DP models, we find that in DP generation when column shuffling is enabled, rejection sampling significantly increases the time required to generate data. In contrast, imputation is more efficient in this scenario.

# C Metrics

## C.1 Perplexity

The perplexity metric serves as a fundamental guage for assessing the performance of language models. It quantifies the uniformity of the model's predictions across a predefined set of tokens in a corpus. Specifically, perplexity is

defined as the exponentiation of the average negative log-likelihood of a sequence, encapsulating the model's ability to predict the next token in a sequence accurately. This measurement reflect how well a model understands the structure and patterns of the language, with lower values indicating higher predictive accuracy and a better grasp of the language nuances.

$$\text{PPL} = \exp\left\{-\frac{1}{t}\sum_i^t \log p(\boldsymbol{t})\right\} \tag{13}$$

where $t$ is the number of total sentences in a corpus and $p(\boldsymbol{t})$ is defined in Equation 1.

*Intuitively,* perplexity is often interpreted as the "effective number of choices" the model is making e.g., a perplexity of 1.8 suggests that the model, on average, has narrowed down the next token to almost 2 equally likely possibilities.

- High Perplexity: Indicates that the model is uncertain about its predictions, implying that the model has not learned well and is making a lot of mistakes.

- Low Perplexity: Suggests that the model is confident in its predictions and is performing well, predicting the next token accurately.

### C.1.1 Disentangled key-value perplexity

The *Key, Value, Other* perplexity are computed by masking out the relevant token perplexity and averaging across the per-example tokens and entire dataset.

## C.2 Tabular-based Metrics

The tabular-based metrics evaluates the synthetic tabular data against the real tabular data.

### C.2.1 Machine Learning Efficacy

The effectiveness of synthetic data is typically assessed through its utility in downstream tasks, aiming to parallel the performance achieved with real data. This evaluation process entails training machine learning models using real data and subsequently evaluating their performance when trained on synthetic data, with comparison made against a reserved set of test data.

### C.2.2 Normalized Histogram Intersection

The normalized histogram intersection which is also referred to as total variation distance measures how aligned the marginal distributions of each column in the generated sample is with the real test data marginal distribution. It provides a quantitative analysis of one-dimensional data distributions by calculating the sum of the minimum probability values across corresponding bins in the real and synthetic data columns. This sum is the averaged over all columns in the dataset, offering a measure of the normalized intersection between the marginal probability distributions of real and synthetic data.

$$\text{Hist}(\boldsymbol{p}_i, \boldsymbol{q}_i) = \sum_c \min(p_c, q_c) \tag{14}$$

$$\text{HI} = \frac{1}{d}\sum_i \text{Hist}(\boldsymbol{p}_i, \boldsymbol{q}_i) \tag{15}$$

where $p_c = \frac{s_c}{|\mathcal{D}|\Delta_i}$ and $q_c = \frac{t_c}{|\mathcal{S}|\Delta_i}$. $\boldsymbol{p}_i$ and $\boldsymbol{q}_i$ represents the histogram probabilities of real ($\mathcal{D}$) and synthetic ($\mathcal{S}$) datasets for feature $i$, respectively. The terms $p_c$ and $q_c$ represent the proportions of category $c$ for feature $i$, with $s_c$ and $t_c$ denoting the counts of real and synthetic samples in category $c$, respectively. The factor $\Delta_i$ is introduced as a normalization term, specifying the bin size for numerical features. The HI is an average of the histogram intersection scores across all features, proving insight into the similarity between the real and synthetic data distributions.

### C.2.3 Pairwise Correlation Similarity Accuracy (CorAcc)

We evaluate the correlation between data columns using the approach described by Tao et al. (2021) and Afonja et al. (2023). Specifically, we use Cramer's V with bias correction for categorical columns, the Correlation Ratio for numerical-categorical columns, and the Pearson Correlation Coefficient (absolute values) for numerical columns. The ranges for these measures are as follows: Cramer's V and Correlation Ratio are bounded between 0 and 1, while the

Pearson Correlation Coefficient spans -1 to 1. Following Tao et al. (2021), correlation values are discretized into four levels: low [0, 0.1), weak [0.1, 0.3), medium [0.3, 0.5), and strong [0.5, 1). The *CorAcc* metric quantifies the similarity between synthetic and original data by measuring the fraction of column pairs where the assigned correlation levels match.

### C.2.4 Pairwise Attribute Distribution Similarity (Pair)

This metric extends the Normalized Histogram Intersection (HIST) by calculating the histogram intersection for all two-way marginals and averaging the results across all attribute pairs. For numerical columns, we discretize the values into bins of size 20 and 50 before computing the intersections.

## D  Setup and Dataset

### D.1  Datasets

**Texas Dataset.**  The Texas Hospital Discharge dataset[12] is a large public use data file provided by the Texas Department of State Health Services. We used the preprocessed version which consists of 100,000 records uniformly selected from a pre-processed file containing patient data from 2013[13] version from Stadler et al. (2022). We retain 18 attributes and assume a binary classification task by predicting only minor and major mortality risk following the setup of Afonja et al. (2023). Duplicates where also removed. The final size of the dataset was therefore reduced to 75,105 which was split to non-overlapping train/test/validation. Validation size is fixed to 1000 for all dataset.

No additional preprocessing was done for Adult and Airline other than removing duplicates.

**List of Column names:**

1. **Adult Income :** *Age, Work Class, FNLWGT, Education, Education Number, Marital Status, Occupation, Relationship, Race, Sex, Capital Gain, Capital Loss, Hours per Week, Native Country,* and *Income*

2. **Airline Passenger Satisfaction:** *ID, Gender, Customer Type, Age, Type of Travel, Class, Flight Distance, Inflight Wi-Fi Service, Departure/Arrival Time Convenience, Ease of Online Booking, Gate Location, Food and Drink, Online Boarding, Seat Comfort, Inflight Entertainment, Onboard Service, Leg Room Service, Baggage Handling, Check-in Service, Inflight Service, Cleanliness, Departure Delay (minutes), Arrival Delay (minutes),* and *Satisfaction (Neutral or Dissatisfied, Satisfied).*

3. **Texas:** *Discharge, Type of Admission, Patient State, Patient Status, Sex Code, Race, Ethnicity, Admission Weekday, Patient Age, Illness Severity, Length of Stay, Total Charges, Total Non-Covered Charges, Total Charges for Accommodation, Total Non-Covered Charges for Accommodation, Total Charges for Ancillary Services, Total Non-Covered Charges for Ancillary Services,* and *Risk of Mortality.*

Table 1 provides statistics of the train-test split, as well as the number of numerical, and categorical columns in the dataset.

## E  Sampling Time

We report the sampling time for generating one dataset of synthetic data. The size of the synthethic data is the same as the training data. The result is shown in  Table6.

## F  Additional Results

### F.1  Comparison of Different Privacy Budget

Table 7 shows the DP result for a higher privacy budget $\varepsilon = 8$. Relaxing the privacy budget shows improved performance for  DP-2Stage across both datasets. Scaling DP-GAN to higher $\varepsilon$ values for the Airline dataset proved challenging, requiring up to five days to run before the process was terminated.

---

[12] https://www.dshs.texas.gov/thcic/
[13] https://github.com/spring-epfl/synthetic_data_release/blob/master/data/texas.csv

| Dataset | Shuffle | DP-Standard | DP-2Stage-O | DP-2Stage-U |
|---------|---------|-------------|-------------|-------------|
| Adult | ✗ | 10 mins | - | - |
| $\varepsilon = \infty$ | ✓ | 10 mins | - | - |
| $\varepsilon = 1$ | ✗ | 11 mins | 11 mins | 10 mins |
| | ✓ | 5 hrs | 6 hrs | 14 mins |
| Airline | ✗ | 1.1 hrs | - | - |
| $\varepsilon = \infty$ | ✓ | 1.6 hrs | - | - |
| $\varepsilon = 1$ | ✗ | 1.1 hrs | 1.1 hrs | 1.2 hrs |
| | ✓ | 42 hrs | 51 hrs | 1.7 hrs |

Table 6: **Sampling Cost**. The synthetic dataset matches the size of the training dataset. ✓ indicates settings with shuffle enabled, while ✗ represents shuffle disabled. The reported values correspond to a single model run and the generation of one synthetic dataset. For Adult, DP-2Stage-O uses Airline as pseudo data and vice-versa.

| | | Adult | | | | Airline | | | |
|---|---|---|---|---|---|---|---|---|---|
| | Method | F1 | AUC | ACC | HIST | F1 | AUC | ACC | HIST |
| $\varepsilon = 1$, $\delta = 10^{-5}$ | *Non-LLM* | | | | | | | | |
| | DP-GAN | $\underline{33.5}_{\pm 20}$ | $\underline{67.7}_{\pm 9}$ | $64.2_{\pm 10}$ | $63.7_{\pm 3}$ | $40.2_{\pm 24}$ | $63.9_{\pm 13}$ | $59.8_{\pm 6}$ | $44.7_{\pm 12}$ |
| | DP-CTGAN | $\mathbf{42.2}_{\pm 20}$ | $\mathbf{78.0}_{\pm 7}$ | $\mathbf{75.7}_{\pm 3}$ | $75.7_{\pm 2}$ | $\underline{67.1}_{\pm 8}$ | $\underline{76.7}_{\pm 8}$ | $\underline{68.0}_{\pm 6}$ | $78.7_{\pm 2}$ |
| | DP-VAE | $0.0_{\pm 0}$ | $50.0_{\pm 0}$ | $\underline{75.6}_{\pm 0}$ | $61.8_{\pm 2}$ | $26.5_{\pm 28}$ | $57.9_{\pm 13}$ | $57.3_{\pm 6}$ | $41.8_{\pm 1}$ |
| | *GPT-2* | | | | | | | | |
| | DP-Standard | $27.8_{\pm 15}$ | $58.5_{\pm 7}$ | $65.2_{\pm 9}$ | $85.7_{\pm 2}$ | $60.5_{\pm 7}$ | $65.3_{\pm 9}$ | $62.4_{\pm 7}$ | $90.3_{\pm 3}$ |
| | DP-2Stage-U | $21.2_{\pm 12}$ | $48.9_{\pm 6}$ | $61.9_{\pm 13}$ | $\underline{86.7}_{\pm 1}$ | $\mathbf{68.5}_{\pm 9}$ | $\mathbf{77.8}_{\pm 10}$ | $\mathbf{72.1}_{\pm 7}$ | $\underline{90.7}_{\pm 1}$ |
| | DP-2Stage-O | $30.4_{\pm 17}$ | $61.6_{\pm 8}$ | $66.7_{\pm 8}$ | $\mathbf{88.5}_{\pm 1}$ | $55.2_{\pm 18}$ | $62.5_{\pm 19}$ | $60.0_{\pm 16}$ | $\mathbf{92.5}_{\pm 1}$ |
| $\varepsilon = 8$, $\delta = 10^{-5}$ | *Non-LLM* | | | | | | | | |
| | DP-GAN | $19.6_{\pm 20}$ | $50.0_{\pm 0}$ | $50.0_{\pm 26}$ | $33.3_{\pm 9}$ | - | - | - | - |
| | DP-CTGAN | $\mathbf{46.5}_{\pm 18}$ | $\mathbf{79.4}_{\pm 4}$ | $\underline{73.1}_{\pm 6}$ | $80.0_{\pm 2}$ | $\underline{67.7}_{\pm 4}$ | $\underline{76.7}_{\pm 5}$ | $67.7_{\pm 4}$ | $76.8_{\pm 1}$ |
| | DP-VAE | $0.0_{\pm 0}$ | $50.0_{\pm 0}$ | $\mathbf{75.6}_{\pm 0}$ | $62.1_{\pm 1}$ | $51.9_{\pm 25}$ | $72.4_{\pm 10}$ | $67.2_{\pm 7}$ | $40.0_{\pm 1}$ |
| | *GPT-2* | | | | | | | | |
| | DP-Standard | $31.3_{\pm 15}$ | $62.2_{\pm 7}$ | $67.7_{\pm 7}$ | $84.5_{\pm 1}$ | $64.9_{\pm 6}$ | $69.8_{\pm 9}$ | $65.9_{\pm 7}$ | $89.8_{\pm 3}$ |
| | DP-2Stage-U | $22.4_{\pm 15}$ | $51.8_{\pm 8}$ | $63.7_{\pm 11}$ | $\underline{86.9}_{\pm 1}$ | $\mathbf{71.9}_{\pm 9}$ | $\mathbf{80.7}_{\pm 10}$ | $\mathbf{74.9}_{\pm 8}$ | $\underline{90.4}_{\pm 1}$ |
| | DP-2Stage-O | $\underline{33.4}_{\pm 16}$ | $\underline{63.8}_{\pm 9}$ | $68.2_{\pm 7}$ | $\mathbf{87.9}_{\pm 1}$ | $64.2_{\pm 11}$ | $71.7_{\pm 10}$ | $\underline{67.8}_{\pm 8}$ | $\mathbf{92.3}_{\pm 1}$ |

Table 7: **DP Benchmark for $\varepsilon = 8$.** Utility metrics (F1, AUC, and ACC) are presented as the averages of logistic regression and XGBoost performance. HIST represents the average histogram intersection scores calculated using bins of 20 and 50. Results are averaged across five model runs and four synthetic datasets per run with standard deviation reported after $\pm$. The best value per column for each $\varepsilon$ is shown in **bold** while second best value is underlined.

## F.2 Marginal-based DP Baseline

**AIM.** Proposed by (McKenna et al., 2022), AIM is a marginal-based model for generating differentially private synthetic data. It is a workload-adaptive algorithm that follows a three-step process: selecting a set of queries, privately measuring those queries, and generating synthetic data from the noisy measurements. AIM employs innovative techniques to iteratively prioritize the most useful measurements, considering both their relevance to the workload and their importance in approximating the input data.

**MST.** Proposed by (McKenna et al., 2021), MST was the winning mechanism of the 2018 NIST Differential Privacy Synthetic Data Competition. It is a general approach for differentially private synthetic data generation that follows

three main steps: (1) selecting a collection of low-dimensional marginals, (2) measuring these marginals using a noise addition mechanism, and (3) generating synthetic data that accurately preserves the measured marginals.

Table 8 presents the results for DP methods, including two marginal-based approaches. The Marginal-based methods shows better performance across most metrics, except for HIST and Pair, where their performance is subpar on the Airline dataset. This is likely due to the higher number of numerical columns in this dataset (see Table 1).

### F.3  Individual Metrics Scores

Table 9 and 10 presents the results for the machine learning models evaluated: XGBoost (XGB) and Logistic Regression (LR). Histogram Intersection score (HIST) is reported for two bin sizes: 20 and 50. The averaged values are summarized in Table 3.

| Dataset | Method | F1 | AUC | ACC | CorAcc | Pair | HIST |
|---|---|---|---|---|---|---|---|
| **Adult** | *Marginal* | | | | | | |
| $\varepsilon = 1$, | AIM | **59.6**$_{\pm 6}$ | **86.8**$_{\pm 1}$ | **80.3**$_{\pm 2}$ | **86.4**$_{\pm 1}$ | **77.1**$_{\pm 9}$ | **88.4**$_{\pm 5}$ |
| $\delta = 10^{-5}$ | MST | 39.6$_{\pm 19}$ | 76.8$_{\pm 1}$ | 72.8$_{\pm 2}$ | 70.0$_{\pm 1}$ | 74.6$_{\pm 10}$ | 87.0$_{\pm 5}$ |
| | | | | | | | |
| | *Non-LLM* | | | | | | |
| | DP-GAN | 33.5$_{\pm 20}$ | 67.7$_{\pm 9}$ | 64.2$_{\pm 10}$ | 39.9$_{\pm 3}$ | 41.2$_{\pm 4}$ | 63.7$_{\pm 3}$ |
| | DP-CTGAN | **42.2**$_{\pm 20}$ | **78.0**$_{\pm 7}$ | **75.7**$_{\pm 3}$ | **51.3**$_{\pm 3}$ | **59.2**$_{\pm 2}$ | **75.7**$_{\pm 2}$ |
| | DP-VAE | 0.0$_{\pm 0}$ | 50.0$_{\pm 0}$ | 75.6$_{\pm 0}$ | 48.8$_{\pm 1}$ | 40.3$_{\pm 1}$ | 61.8$_{\pm 2}$ |
| | | | | | | | |
| | *GPT-2* | | | | | | |
| | DP-Standard | 27.8$_{\pm 15}$ | 58.5$_{\pm 7}$ | 65.2$_{\pm 9}$ | 55.0$_{\pm 1}$ | 68.4$_{\pm 1}$ | 85.7$_{\pm 2}$ |
| | DP-2Stage-U | 21.2$_{\pm 12}$ | 48.9$_{\pm 6}$ | 61.9$_{\pm 13}$ | 55.0$_{\pm 1}$ | **76.1**$_{\pm 1}$ | 86.7$_{\pm 1}$ |
| | DP-2Stage-O | | | | | | |
| | +airline | 30.4$_{\pm 17}$ | **61.6**$_{\pm 8}$ | **66.7**$_{\pm 8}$ | 55.4$_{\pm 1}$ | 72.3$_{\pm 1}$ | **88.5**$_{\pm 1}$ |
| | +texas | **31.6**$_{\pm 13}$ | 60.5$_{\pm 7}$ | 66.4$_{\pm 8}$ | **55.6**$_{\pm 1}$ | 71.3$_{\pm 1}$ | 86.9$_{\pm 1}$ |
| **Airline** | *Marginal* | | | | | | |
| $\varepsilon = 1$, | AIM | **77.3**$_{\pm 5}$ | **88.9**$_{\pm 4}$ | **78.2**$_{\pm 5}$ | **91.8**$_{\pm 1}$ | 46.7$_{\pm 3}$ | 68.1$_{\pm 2}$ |
| $\delta = 10^{-5}$ | MST | 72.2$_{\pm 6}$ | 83.4$_{\pm 5}$ | 75.2$_{\pm 4}$ | 72.7$_{\pm 0}$ | 46.3$_{\pm 3}$ | **68.2**$_{\pm 2}$ |
| | | | | | | | |
| | *Non-LLM* | | | | | | |
| | DP-GAN | 40.2$_{\pm 24}$ | 63.9$_{\pm 13}$ | 59.8$_{\pm 6}$ | 37.4$_{\pm 9}$ | 22.2$_{\pm 13}$ | 44.7$_{\pm 12}$ |
| | DP-CTGAN | **67.1**$_{\pm 8}$ | **76.7**$_{\pm 8}$ | **68.0**$_{\pm 6}$ | 31.7$_{\pm 2}$ | **62.2**$_{\pm 2}$ | **78.7**$_{\pm 2}$ |
| | DP-VAE | 26.5$_{\pm 28}$ | 57.9$_{\pm 13}$ | 57.3$_{\pm 6}$ | **46.6**$_{\pm 1}$ | 20.6$_{\pm 0}$ | 41.8$_{\pm 1}$ |
| | | | | | | | |
| | *GPT-2* | | | | | | |
| | DP-Standard | 60.5$_{\pm 7}$ | 65.3$_{\pm 9}$ | 62.4$_{\pm 7}$ | 64.0$_{\pm 2}$ | 77.0$_{\pm 2}$ | 90.3$_{\pm 3}$ |
| | DP-2Stage-U | **68.5**$_{\pm 9}$ | **77.8**$_{\pm 10}$ | **72.1**$_{\pm 7}$ | 65.3$_{\pm 1}$ | **80.8**$_{\pm 1}$ | 90.7$_{\pm 1}$ |
| | DP-2Stage-O | | | | | | |
| | +adult | 55.2$_{\pm 18}$ | 62.5$_{\pm 19}$ | 60.0$_{\pm 16}$ | **66.8**$_{\pm 1}$ | 80.1$_{\pm 1}$ | **92.5**$_{\pm 1}$ |
| | +texas | 52.5$_{\pm 13}$ | 61.0$_{\pm 13}$ | 58.4$_{\pm 10}$ | 66.1$_{\pm 2}$ | 78.7$_{\pm 2}$ | 90.4$_{\pm 1}$ |
| **Texas** | *Marginal* | | | | | | |
| $\varepsilon = 1$, | AIM | **84.5**$_{\pm 1}$ | 98.3$_{\pm 0}$ | **94.2**$_{\pm 1}$ | **81.0**$_{\pm 2}$ | 93.0$_{\pm 5}$ | 98.9$_{\pm 0}$ |
| $\delta = 10^{-5}$ | MST | 81.7$_{\pm 0}$ | 94.8$_{\pm 0}$ | 93.2$_{\pm 0}$ | 77.0$_{\pm 0}$ | **97.5**$_{\pm 0}$ | **99.0**$_{\pm 0}$ |
| | | | | | | | |
| | *Non-LLM* | | | | | | |
| | DP-GAN | 13.7$_{\pm 16}$ | 58.3$_{\pm 12}$ | 78.3$_{\pm 8}$ | 36.1$_{\pm 7}$ | 34.6$_{\pm 5}$ | 68.3$_{\pm 7}$ |
| | DP-CTGAN | **63.9**$_{\pm 11}$ | **91.4**$_{\pm 3}$ | 82.6$_{\pm 19}$ | 43.9$_{\pm 6}$ | **66.9**$_{\pm 6}$ | **84.7**$_{\pm 3}$ |
| | DP-VAE | 0.0$_{\pm 0}$ | 50.0$_{\pm 0}$ | 82.5$_{\pm 0}$ | **62.1**$_{\pm 1}$ | 43.9$_{\pm 1}$ | 77.9$_{\pm 1}$ |
| | | | | | | | |
| | *GPT-2* | | | | | | |
| | DP-Standard | 55.4$_{\pm 10}$ | 90.2$_{\pm 5}$ | 77.1$_{\pm 11}$ | 70.3$_{\pm 1}$ | 60.6$_{\pm 1}$ | 92.3$_{\pm 1}$ |
| | DP-2Stage-U | 23.5$_{\pm 14}$ | 59.8$_{\pm 14}$ | 67.3$_{\pm 17}$ | 68.2$_{\pm 0}$ | **80.7**$_{\pm 6}$ | **93.4**$_{\pm 0}$ |
| | DP-2Stage-O | | | | | | |
| | +adult | **74.8**$_{\pm 4}$ | **96.7**$_{\pm 1}$ | **89.1**$_{\pm 3}$ | 69.9$_{\pm 2}$ | 60.5$_{\pm 2}$ | 91.7$_{\pm 1}$ |
| | +airline | 74.3$_{\pm 5}$ | 96.4$_{\pm 0}$ | 88.8$_{\pm 3}$ | **70.9**$_{\pm 1}$ | 62.0$_{\pm 2}$ | 93.2$_{\pm 0}$ |

Table 8: **DP Benchmark.** Utility metrics (F1, AUC, and ACC) are presented as the averages of two ML Models (XGBoost and Logistic Regression). **Pair**, and **Hist** are reported as averages of two bin sizes (Bins 20 and 50). Results are averaged across five model runs and four synthetic datasets per run with standard deviation reported after $\pm$. The best value per method group for $\varepsilon = 1$ is shown in **bold**.

| Dataset | Method | XGB (F1) | XGB (AUC) | XGB (ACC) | LR (F1) | LR (AUC) | LR (ACC) | HIST (bin=50) | HIST (bin=20) |
|---|---|---|---|---|---|---|---|---|---|
| **Adult** | *Marginal* | | | | | | | | |
| $\varepsilon = 1$ | AIM | $\mathbf{54.0}_{\pm4}$ | $\mathbf{86.3}_{\pm1}$ | $\mathbf{81.8}_{\pm0}$ | $\mathbf{65.2}_{\pm0}$ | $\mathbf{87.3}_{\pm0}$ | $\mathbf{78.8}_{\pm1}$ | $\mathbf{83.5}_{\pm0}$ | $\mathbf{93.3}_{\pm1}$ |
| $\delta = 10^{-5}$ | MST | $20.5_{\pm3}$ | $76.7_{\pm1}$ | $74.5_{\pm0}$ | $58.7_{\pm0}$ | $77.0_{\pm2}$ | $71.0_{\pm0}$ | $81.8_{\pm2}$ | $92.2_{\pm0}$ |
| | *Non-LLM* | | | | | | | | |
| | DP-GAN | $27.4_{\pm24}$ | $67.6_{\pm10}$ | $69.1_{\pm9}$ | $39.6_{\pm14}$ | $67.8_{\pm9}$ | $59.4_{\pm8}$ | $61.9_{\pm2}$ | $65.5_{\pm3}$ |
| | DP-CTGAN | $\mathbf{38.6}_{\pm21}$ | $\mathbf{77.2}_{\pm7}$ | $\mathbf{76.0}_{\pm3}$ | $\mathbf{45.8}_{\pm19}$ | $\mathbf{78.8}_{\pm7}$ | $75.3_{\pm3}$ | $\mathbf{75.0}_{\pm2}$ | $\mathbf{76.4}_{\pm2}$ |
| | DP-VAE | $0.0_{\pm0}$ | $50.0_{\pm0}$ | $75.6_{\pm0}$ | $0.0_{\pm0}$ | $50.0_{\pm0}$ | $\mathbf{75.6}_{\pm0}$ | $60.1_{\pm0}$ | $63.5_{\pm0}$ |
| | *GPT-2* | | | | | | | | |
| | DP-Standard | $13.9_{\pm7}$ | $55.5_{\pm6}$ | $73.0_{\pm1}$ | $41.6_{\pm5}$ | $61.4_{\pm7}$ | $57.4_{\pm6}$ | $85.1_{\pm2}$ | $86.2_{\pm2}$ |
| | DP-2Stage-U | $10.3_{\pm5}$ | $48.4_{\pm4}$ | $\mathbf{74.5}_{\pm1}$ | $32.1_{\pm6}$ | $49.3_{\pm8}$ | $49.4_{\pm4}$ | $86.3_{\pm1}$ | $87.1_{\pm1}$ |
| | DP-2Stage-O | | | | | | | | |
| | +airline | $\mathbf{15.0}_{\pm7}$ | $\mathbf{55.9}_{\pm6}$ | $73.5_{\pm1}$ | $\mathbf{45.9}_{\pm5}$ | $\mathbf{67.3}_{\pm6}$ | $\mathbf{59.8}_{\pm5}$ | $\mathbf{88.2}_{\pm1}$ | $\mathbf{88.8}_{\pm1}$ |
| | +texas | $9.6_{\pm7}$ | $56.7_{\pm5}$ | $73.4_{\pm1}$ | $43.6_{\pm5}$ | $64.4_{\pm5}$ | $59.4_{\pm5}$ | $86.5_{\pm1}$ | $87.3_{\pm1}$ |
| **Airline** | *Marginal* | | | | | | | | |
| $\varepsilon = 1$ | AIM | $\mathbf{73.0}_{\pm4}$ | $\mathbf{85.2}_{\pm3}$ | $\mathbf{74.3}_{\pm4}$ | $\mathbf{81.7}_{\pm0}$ | $\mathbf{92.7}_{\pm0}$ | $\mathbf{82.1}_{\pm0}$ | $65.9_{\pm0}$ | $70.2_{\pm0}$ |
| $\delta = 10^{-5}$ | MST | $69.2_{\pm8}$ | $80.4_{\pm6}$ | $74.0_{\pm5}$ | $75.2_{\pm0}$ | $86.3_{\pm0}$ | $76.4_{\pm0}$ | $\mathbf{66.1}_{\pm0}$ | $\mathbf{70.3}_{\pm0}$ |
| | *Non-LLM* | | | | | | | | |
| | DP-GAN | $38.3_{\pm25}$ | $65.1_{\pm14}$ | $60.3_{\pm6}$ | $42.1_{\pm24}$ | $62.7_{\pm12}$ | $59.2_{\pm6}$ | $43.4_{\pm12}$ | $46.0_{\pm12}$ |
| | DP-CTGAN | $\mathbf{64.1}_{\pm10}$ | $\mathbf{74.0}_{\pm8}$ | $\mathbf{65.9}_{\pm6}$ | $\mathbf{70.1}_{\pm5}$ | $\mathbf{79.4}_{\pm7}$ | $\mathbf{70.1}_{\pm5}$ | $\mathbf{78.5}_{\pm2}$ | $\mathbf{79.0}_{\pm1}$ |
| | DP-VAE | $1.0_{\pm3}$ | $54.4_{\pm11}$ | $56.7_{\pm1}$ | $52.0_{\pm14}$ | $61.4_{\pm14}$ | $58.0_{\pm8}$ | $41.5_{\pm0}$ | $42.2_{\pm0}$ |
| | *GPT-2* | | | | | | | | |
| | DP-Standard | $55.8_{\pm3}$ | $62.1_{\pm7}$ | $60.0_{\pm6}$ | $65.1_{\pm6}$ | $68.4_{\pm10}$ | $64.9_{\pm8}$ | $89.6_{\pm4}$ | $91.1_{\pm3}$ |
| | DP-2Stage-U | $\mathbf{64.5}_{\pm10}$ | $\mathbf{73.8}_{\pm9}$ | $\mathbf{70.0}_{\pm7}$ | $\mathbf{72.5}_{\pm6}$ | $\mathbf{81.8}_{\pm8}$ | $\mathbf{74.1}_{\pm6}$ | $90.1_{\pm0}$ | $91.3_{\pm1}$ |
| | DP-2Stage-O | | | | | | | | |
| | +adult | $53.5_{\pm16}$ | $61.8_{\pm16}$ | $59.2_{\pm13}$ | $56.9_{\pm20}$ | $63.3_{\pm22}$ | $60.8_{\pm18}$ | $\mathbf{92.0}_{\pm1}$ | $\mathbf{93.0}_{\pm1}$ |
| | +airline | $48.8_{\pm12}$ | $58.0_{\pm12}$ | $57.5_{\pm9}$ | $56.2_{\pm14}$ | $64.0_{\pm13}$ | $59.2_{\pm11}$ | $89.8_{\pm1}$ | $91.0_{\pm1}$ |
| **Texas** | *Marginal* | | | | | | | | |
| $\varepsilon = 1$ | AIM | $\mathbf{85.4}_{\pm1.0}$ | $\mathbf{98.3}_{\pm0.0}$ | $\mathbf{94.9}_{\pm0}$ | $\mathbf{83.5}_{\pm1}$ | $\mathbf{98.4}_{\pm0}$ | $\mathbf{93.4}_{\pm0}$ | $98.7_{\pm0}$ | $\mathbf{99.2}_{\pm0}$ |
| $\delta = 10^{-5}$ | MST | $81.3_{\pm0}$ | $94.7_{\pm0}$ | $93.2_{\pm0}$ | $82.0_{\pm0}$ | $94.9_{\pm1}$ | $93.3_{\pm0}$ | $\mathbf{98.8}_{\pm0}$ | $\mathbf{99.2}_{\pm0}$ |
| | *Non-LLM* | | | | | | | | |
| | DP-GAN | $13.4_{\pm19}$ | $61.2_{\pm12}$ | $78.2_{\pm9}$ | $14.1_{\pm14}$ | $55.5_{\pm12}$ | $78.5_{\pm8}$ | $66.0_{\pm7}$ | $70.5_{\pm6}$ |
| | DP-CTGAN | $\mathbf{60.3}_{\pm15}$ | $\mathbf{91.2}_{\pm3}$ | $76.4_{\pm25}$ | $\mathbf{67.5}_{\pm2}$ | $\mathbf{91.6}_{\pm2}$ | $\mathbf{88.7}_{\pm1}$ | $\mathbf{83.9}_{\pm3}$ | $\mathbf{85.5}_{\pm3}$ |
| | DP-VAE | $0.0_{\pm0}$ | $50.0_{\pm0}$ | $\mathbf{82.5}_{\pm0}$ | $0.0_{\pm0}$ | $50.0_{\pm0}$ | $82.5_{\pm0}$ | $77.0_{\pm0}$ | $78.9_{\pm0}$ |
| | *GPT-2* | | | | | | | | |
| | DP-Standard | $58.5_{\pm12}$ | $88.8_{\pm5}$ | $86.1_{\pm2}$ | $52.2_{\pm7}$ | $91.5_{\pm5}$ | $68.2_{\pm9}$ | $92.2_{\pm1}$ | $92.4_{\pm0}$ |
| | DP-2Stage-U | $11.3_{\pm8}$ | $51.1_{\pm10}$ | $82.5_{\pm1}$ | $36.2_{\pm6}$ | $68.5_{\pm12}$ | $52.0_{\pm10}$ | $\mathbf{93.2}_{\pm0}$ | $\mathbf{93.6}_{\pm0}$ |
| | DP-2Stage-O | | | | | | | | |
| | +adult | $77.6_{\pm2}$ | $\mathbf{96.5}_{\pm1}$ | $91.1_{\pm1}$ | $\mathbf{72.0}_{\pm4}$ | $\mathbf{96.9}_{\pm1}$ | $\mathbf{87.0}_{\pm3}$ | $91.3_{\pm1}$ | $92.1_{\pm1}$ |
| | +airline | $\mathbf{78.0}_{\pm2}$ | $96.3_{\pm0}$ | $\mathbf{91.3}_{\pm1}$ | $70.6_{\pm3}$ | $96.4_{\pm0}$ | $86.3_{\pm2}$ | $93.0_{\pm0}$ | $93.4_{\pm0}$ |

Table 9: **DP Benchmarks showing individual metric result.** For each dataset, the best value per method group for $\varepsilon = 1$ is shown in **bold**. Results are averaged across five model runs with varying random seeds, with four synthetic datasets generated per run. LR refers to the Logistic Regression model, and XGB represents the XGBoost model.

| | Dataset | | XGB (F1) | XGB (AUC) | XGB (ACC) | LR (F1) | LR (AUC) | LR (ACC) | HIST (bin=50) | HIST (bin=20) |
|---|---|---|---|---|---|---|---|---|---|---|
| $\varepsilon = 8,$ | Adult | DP-Standard | $17.3_{\pm6}$ | $\mathbf{58.6}_{\pm4}$ | $73.6_{\pm1}$ | $45.2_{\pm6}$ | $65.8_{\pm8}$ | $61.8_{\pm4}$ | $84.1_{\pm1}$ | $84.9_{\pm1}$ |
| $\delta = 10^{-5}$ | | DP-2Stage-U | $9.7_{\pm6}$ | $49.4_{\pm4}$ | $\mathbf{74.4}_{\pm1}$ | $35.1_{\pm8}$ | $54.1_{\pm9}$ | $52.9_{\pm5}$ | $86.4_{\pm1}$ | $87.4_{\pm1}$ |
| | | DP-2Stage-O | $\mathbf{19.1}_{\pm9}$ | $58.5_{\pm9}$ | $74.2_{\pm1}$ | $\mathbf{47.6}_{\pm5}$ | $\mathbf{69.1}_{\pm7}$ | $\mathbf{62.3}_{\pm4}$ | $\mathbf{87.5}_{\pm1}$ | $\mathbf{88.3}_{\pm1}$ |
| | Airline | DP-Standard | $61.4_{\pm4}$ | $66.1_{\pm7}$ | $63.4_{\pm7}$ | $68.5_{\pm6}$ | $73.6_{\pm9}$ | $68.4_{\pm7}$ | $89.1_{\pm3}$ | $90.5_{\pm3}$ |
| | | DP-2Stage-U | $\mathbf{69.1}_{\pm11}$ | $\mathbf{77.8}_{\pm10}$ | $\mathbf{73.6}_{\pm8}$ | $\mathbf{74.7}_{\pm7}$ | $\mathbf{83.5}_{\pm10}$ | $\mathbf{76.2}_{\pm8}$ | $89.9_{\pm1}$ | $90.9_{\pm1}$ |
| | | DP-2Stage-O | $60.2_{\pm8}$ | $67.9_{\pm6}$ | $65.2_{\pm5}$ | $68.2_{\pm11}$ | $75.5_{\pm13}$ | $70.3_{\pm10}$ | $\mathbf{91.9}_{\pm1}$ | $\mathbf{92.8}_{\pm1}$ |

Table 10: **DP-GPT-2 Benchmarks showing individual metric result for $\varepsilon = 8$.** For each dataset, the best value corresponding to different privacy budgets ($\varepsilon$) is highlighted in **bold**. Results are averaged across five model runs with varying random seeds, with four synthetic datasets generated per run. LR refers to the Logistic Regression model, and XGB represents the XGBoost model.