# OpenReview forum: "DP-2Stage: Adapting Language Models as Differentially Private Tabular Data Generators"
_TMLR — Accepted by TMLR_

### Review · Reviewer_5EZR · 2024-12-18

**Summary Of Contributions:**

This paper proposes two methods for synthetic data generation via fine-tuning language models. Differentially private fine-tuning is performed in two steps: (1) fine-tune the LM on uniformly generated or public data, then (2) fine-tune with DP-SGD on private data.

**Audience:**

Yes

**Broader Impact Concerns:**

The broader impact provided by the authors is sufficient.

**Claims And Evidence:**

Yes

**Requested Changes:**

- Please include exact definitions of Standard-DP
- Please include at least one metric to evaluate the quality of synthetic data
- For the experiment, can the authors comment on whether the Adult and Airline datasets are already parts of the training set of GPT-2, and how this fact might affect the results of the experiments in this paper.
- From Section 5.3 onwards, it should be emphasized in the tables that $\delta=10^{-5}$.
- Probably very minor, but I think $1e^{-5}$ has a very different meaning from `1e-5`; the former is $e^{-5}$ with the mathematical constant $e$, while the latter is $10^{-5}$.

I also would like to make the following request. If the experimental results are convincing, I will recommend the paper for publication.
- Could the authors also evaluate the data synthetic methods using pairwise metrics, such as "Pair" and "Corr" in [this paper](https://aaai-ppai22.github.io/files/26.pdf).
- Could the authors also evaluate previous DP-focused tabular data synthetic methods such as [DPSyn](https://github.com/agl-c/deid2_dpsyn), [PrivMRF](https://github.com/caicre/PrivMRF) and [Private-PGM](https://github.com/ryan112358/private-pgm)?

**Edit**: I believe the extensive experiment by the authors provides sufficient contribution in the field, and it shows that GPT2-based method can be competitive against the current best method. For this reason I recommend for acceptance.

**Strengths And Weaknesses:**

## Strengths

- Experimentally, the proposed methods improve over the standard DP fine-tuning.
- The proposed methods improve over other DP synthetic data methods in Normalized Histogram Intersection (Hist).

## Weaknesses

- As I understand, the authors report the classifications metrics at the average of both logistic regression's and XGBoost's scores. As a result, the standard deviations of these scores are quite large and hard to interpret. I think it makes more sense to report the averages and standard deviations of the individual models.
-  Even though all the GPT-2-based methods achieve high Hist, their classification scores are quite low. Maybe the classification metrics are not quite indicative of the quality of the synthetic data? The authors might want to discuss the discrepancies between the classification metrics and Hist.
- I do not know how the authors ensure that the fine-tuning in the first step in DP-2Stage-U

---

> ### Author Response · Authors · 2024-12-20
> **Response to Reviewer 5EZR**
>
> Thank you for your review. Below we provide in-line responses to your questions and concerns.
>
> ### Weaknesses
> - As I understand, the authors report the classifications metrics at the average of both logistic regression's and XGBoost's scores. As a result, the standard deviations of these scores are quite large and hard to interpret. I think it makes more sense to report the averages and standard deviations of the individual models.
> >We have provided a detailed breakdown of each metric for the individual DP GPT-2 models in Appendix E. Additionally, we added a sentence in section 5.3 to refer readers to the appendix. Please let us know if this addresses their concerns.
>
> - Even though all the GPT-2-based methods achieve high Hist, their classification scores are quite low. Maybe the classification metrics are not quite indicative of the quality of the synthetic data? The authors might want to discuss the discrepancies between the classification metrics and Hist.
> >The classification metrics and histogram intersection (Hist) metric serve as complementary evaluations, highlighting different aspects of synthetic data quality. While a high Hist score reflects that the column-wise distributions of the synthetic data closely match those of the real data, it does not necessarily guarantee high classification utility. This discrepancy arises because Hist focuses on preserving marginal distributions, whereas utility metrics assess how well the synthetic data captures feature-target relationships critical for downstream tasks. For instance, generating synthetic data by independently sampling from the marginal distributions of real data can result in a high Hist score but low utility, as feature-target relationships essential for classification tasks may not be preserved. The fidelity-utility tradeoff in private generative models has been studied in other domains, such as synthetic images, where approaches that achieve high fidelity (e.g., preserving pixel-level statistics) often perform poorly on downstream tasks [1] due to their focus on fitting the entire data distribution. To address this challenge, researchers have proposed generative models that prioritize utility, allowing for a slight degradation in fidelity during optimization [1]. We recognize that evaluating synthetic data comprehensively requires a suite of metrics to capture its overall quality, as emphasized in prior work [2, 3, 4]. This need for holistic evaluation frameworks, is further discussed in the limitations and future work sections of the paper.
>
>     >[1] Chen, D., Kerkouche, R. and Fritz, M., 2022. Private set generation with discriminative information. Advances in Neural Information Processing Systems, 35, pp.14678-14690.
>
>     >[2] Dankar, F.K., Ibrahim, M.K. and Ismail, L., 2022. A multi-dimensional evaluation of synthetic data generators. IEEE Access, 10, pp.11147-11158.
>
>     >[3] Afonja, T., Chen, D. and Fritz, M., 2023, September. MargCTGAN: A “Marginally” Better CTGAN for the Low Sample Regime. In DAGM German Conference on Pattern Recognition (pp. 524-537). Cham: Springer Nature Switzerland.
>
>     >[4] Chen, D., Oestreich, M., Afonja, T., Kerkouche, R., Becker, M. and Fritz, M., 2024. Towards biologically plausible and private gene expression data generation. Proceedings on Privacy Enhancing Technologies, 2024.
>
>
> - I do not know how the authors ensure that the fine-tuning in the first step in DP-2Stage-U
> >As outlined in Section 4.1, the fine-tuning process in the first stage of DP-2Stage-U involves creating a pseudo-dataset. This dataset is generated by independently sampling from a uniform distribution based on the parameters of the real data. The pre-trained LLM is then fine-tuned on this pseudo-dataset for 5 epochs. In the second stage, the fine-tuned model from the first stage is further fine-tuned on private data, ensuring differential privacy through DPSGD. We are happy to address any further questions you may have.

---

> > ### Comment · Reviewer_5EZR · 2024-12-20
> > **Thanks for the reply**
> >
> > Thank you for the fast reply. I am quite satisfied with the answers.
> >
> > For the last point in Weaknesses, I wrote only first half of my concern and forgot to complete it. What I'd like to ask is:
> >
> > - How do you ensure that the fine-tuning in the first step in DP-2Stage-U is differentially private? We know that releasing parameters of the sensitive data (inluding maximums and minimums) is not $\epsilon$-DP. Uniform sampling with these parameters is not $\epsilon$-DP either.

---

> > > ### Author Response · Authors · 2025-01-01
> > > **Response to Weakness**
> > >
> > > ### Weaknesses
> > >
> > > - How do you ensure that the fine-tuning in the first step in DP-2Stage-U is differentially private? We know that releasing parameters of the sensitive data (inluding maximums and minimums) is not ϵ-DP. Uniform sampling with these parameters is not ϵ-DP either.
> > > >  We thank the reviewer for this insightful question and hope the reviewer is having a great start into the New Year.
> > > We proposed DP-2Stage-U as a feasible solution in cases where the range of numerical features as well as categorical features are not sensitive information and can be public. However, in the opposite case as also highlighted in [1], we completely agree that such information must also be derived using differential privacy to ensure that the entire solution is differentially private.  Another alternative we proposed, in cases where this information is indeed considered sensitive, is to use a public dataset, even one of a completely different nature (out-of-distribution), to ensure that the solution remains differentially private. We hope our response addressed your concern, and we would be more than happy to provide further clarification if necessary.
> > >
> > >      > [1] Stadler, T., Oprisanu, B. and Troncoso, C., 2022. Synthetic data–anonymisation groundhog day. In 31st USENIX Security Symposium (USENIX Security 22).

---

> > > > ### Author Response · Authors · 2025-01-01
> > > > **Update on Requested Changes**
> > > >
> > > > ### Requested Changes
> > > > I also would like to make the following request. If the experimental results are convincing, I will recommend the paper for publication.
> > > >
> > > > - Could the authors also evaluate the data synthetic methods using pairwise metrics, such as "Pair" and "Corr" in this paper [1]
> > > > - Could the authors also evaluate previous DP-focused tabular data synthetic methods such as DPSyn, PrivMRF and Private-PGM?
> > > > > We have implemented the “Pair” and “Corr” metrics as described in [1] and present the results in Table 1 and 2 of  [this PDF](https://anonymous.4open.science/r/DP-2Stage-B8B1/docs/reviewer1_req.pdf). The metrics were calculated by averaging the outcomes over 5 runs of the algorithm on 4 synthetic datasets, each of the same size as the real dataset. The results demonstrate that our proposed methods outperform in these metrics, achieving higher accuracy in preserving the correlation between column pairs compared to the real test set. Furthermore, our methods yield a higher inverse total variation distance (or normalized histogram intersection score) for the joint distribution of column pairs in the real test dataset. We provide code in [https://anonymous.4open.science/r/DP-2Stage-B8B1/metrics/column_pair.py](https://anonymous.4open.science/r/DP-2Stage-B8B1/metrics/column_pair.py).
> > > >
> > > >     >Regarding the requested changes for additional baselines, we investigated this and observed the following: DPSyn [2] improves upon PGM [3], while PrivMRF [4] builds on top of PGM [3] and Private-PGM [5] improves PGM [3]. Additionally, AIM [6] is a recently proposed method that builds on previous approaches [5] and Its implementation is available through the [SmartNoise SDK](https://github.com/opendp/smartnoise-sdk/tree/main)—a popular framework which we used to benchmark the DP-GAN and DP-CTGAN baselines and is also used in [1].
> > > >
> > > >     > To ensure consistency, we included results for AIM [6] and MST [7] (which utilizes Private-PGM) in Tables 3 and 4 of  [this PDF](https://anonymous.4open.science/r/DP-2Stage-B8B1/docs/reviewer1_req.pdf). These baselines were chosen instead of DPSyn [2], PGM [3], and PrivMRF [4], given their integration with the SmartNoise SDK.
> > > >     >Our results show that Marginal-Based methods achieve superior utility scores across both datasets. However, for the Airline dataset, our methods demonstrate best performance on the “Pair” and “Hist” metrics. We will include these results in the Appendix of the paper for transparency. Please let us know if this addresses your concerns or if further clarification is needed.
> > > >
> > > >     >[1] Tao, Y., McKenna, R., Hay, M., Machanavajjhala, A. and Miklau, G., 2022. Benchmarking differentially private synthetic data generation algorithms. AAAI Workshop on Privacy-Preserving Artificial Intelligence.
> > > >
> > > >     > [2] Zhang, Z., Wang, T., Li, N., Honorio, J., Backes, M., He, S., Chen, J. and Zhang, Y., 2021. {PrivSyn}: Differentially private data synthesis. In 30th USENIX Security Symposium (USENIX Security 21).
> > > >
> > > >     > [3] McKenna, R., Sheldon, D. and Miklau, G., 2019, May. Graphical-model based estimation and inference for differential privacy. In International Conference on Machine Learning. PMLR.
> > > >
> > > >     > [4] Cai, K., Lei, X., Wei, J. and Xiao, X., 2021. Data synthesis via differentially private markov random fields. Proceedings of the VLDB Endowment.
> > > >
> > > >     > [5] McKenna, R., Pradhan, S., Sheldon, D.R. and Miklau, G., 2021. Relaxed marginal consistency for differentially private query answering. Advances in Neural Information Processing Systems.
> > > >
> > > >     > [6] McKenna, R., Mullins, B., Sheldon, D. and Miklau, G., 2022. AIM: an adaptive and iterative mechanism for differentially private synthetic data. Proceedings of the VLDB Endowment.
> > > >
> > > >     > [7] McKenna, R., Miklau, G. and Sheldon, D., 2021. Winning the NIST Contest: A scalable and general approach to differentially private synthetic data. Journal of Privacy and Confidentiality.

---

> > > > > ### Comment · Reviewer_5EZR · 2025-01-02
> > > > > **Response to the update**
> > > > >
> > > > > Thanks for the extensive experiment. It is not surprising that the marginal-based methods perform best in the marginal-based metrics. Nonetheless, the experiment results show that proposed GPT2-based methods can be competitive against those methods. For future research, I believe that these marginal-focused techniques can be added to the pipeline in order to further improve the quality the synthetic data.
> > > > >
> > > > > > we completely agree that such information must also be derived using differential privacy to ensure that the entire solution is differentially private.
> > > > >
> > > > > I believe that this is an important that should be included in the main paper. With sufficiently large datasets, the DP statistics should be close to the real values, so I recommend to apply DP techniques to privatize these values and obtain privacy guarantees.
> > > > >
> > > > > Since the authors have provided sufficient insight on where the LM-based methods stand for DP data synthesis, I recommend the paper for acceptance.

---

> > > > > > ### Author Response · Authors · 2025-01-02
> > > > > > **Thank you**
> > > > > >
> > > > > > We sincerely thank Reviewer 5EZR for their insightful comments and constructive feedback, which has greatly contributed to improving the quality of our paper. We appreciate the recommendation to apply DP techniques to compute private statistics for the DP-2Stage-U method. This is an excellent suggestion that we plan to explore in future work.
> > > > > >
> > > > > > Furthermore, we are grateful for your recognition of our efforts in positioning LM-based methods for DP tabular data synthesis and for recommending our paper for acceptance.

---

> ### Author Response · Authors · 2024-12-20
> **Response to Reviewer 5EZR's Requested Changes (1/2)**
>
> ### Requested Changes:
> - Please include exact definitions of Standard-DP
> >In Section 4, we have provided a concise definition of standard DP as: "Standard DP training, where the LLM model is directly fine-tuned using DPSGD." Additionally we have provided a more detailed description below and will include it in Appendix A.1 and referenced in section 4.
>
>     > \subsection{Standard DP}
>     >We define the \textit{Standard} DP or \textbf{DP-Standard} training process as fine-tuning the LLM directly on private data using the Differentially Private Stochastic Gradient Descent (DPSGD)~\citep{abadi2016deep} mechanism. DPSGD ensures that the training process complies with the formal definition of differential privacy through the following steps: (1) Gradients are computed for each individual sample within a mini-batch. (2) Gradients are clipped to a fixed norm to bound sensitivity. (3) Gaussian noise, calibrated to the privacy parameters ($\varepsilon,\delta$), is added to the aggregated gradients. (4) The resulting noisy gradients are used to update the model's parameters.
>
>
> - Please include at least one metric to evaluate the quality of synthetic data
> >The classification metrics and histogram metrics are widely used in the literature to evaluate the quality of synthetic datasets [1,2,3,4]. We will also include the metrics suggested by the reviewer [5].
>
>     >[1] Xu, L., Skoularidou, M., Cuesta-Infante, A. and Veeramachaneni, K., 2019. Modeling tabular data using conditional gan. Advances in neural information processing systems, 32.
>
>     >[2] Borisov, V., Seßler, K., Leemann, T., Pawelczyk, M. and Kasneci, G., 2023. Language models are realistic tabular data generators. International Conference on Learning Representations.
>
>     >[3] Afonja, T., Chen, D. and Fritz, M., 2023, September. MargCTGAN: A “Marginally” Better CTGAN for the Low Sample Regime. In DAGM German Conference on Pattern Recognition (pp. 524-537). Cham: Springer Nature Switzerland.
>
>     >[4] Kotelnikov, A., Baranchuk, D., Rubachev, I. and Babenko, A., 2023, July. Tabddpm: Modelling tabular data with diffusion models. In International Conference on Machine Learning (pp. 17564-17579). PMLR.
>
>     >[5] Tao, Y., McKenna, R., Hay, M., Machanavajjhala, A. and Miklau, G., 2022. Benchmarking differentially private synthetic data generation algorithms. AAAI Workshop on Privacy-Preserving Artificial Intelligence.
>
> - For the experiment, can the authors comment on whether the Adult and Airline datasets are already part of the training set of GPT-2, and how this fact might affect the results of the experiments in this paper.
> >The GPT-2 model was reported in [1] to have been trained on a dataset known as WebText, constructed by scraping text from over 8 million web pages linked in Reddit posts that had received at least three upvotes prior to December 2017.  Given the Adult dataset's long-standing availability as a popular benchmarking dataset in the UCI Machine Learning Repository and the Airline dataset's presence on Kaggle, it is plausible that these datasets may have been included in the pre-training corpus of GPT-2. However, we do not have explicit knowledge whether the datasets were indeed included as the pre-training dataset of GPT-2 and the curated WebText is not publicly available for inspection. However, regardless of the inclusion, our contributions remain critical: (1) public datasets or pre-trained models have been widely used in many applications [2] (and arguably privacy-preserving). Our work investigates how to adapt language models efficiently under DP settings, which are timely and under exploration in the existing literature [3,4]. (2) Even if the language model was exposed to sensitive data during pre-training, our approach theoretically constrains any further leakage compared to their non-DP counterparts (up to $\varepsilon,\delta$). Nonetheless, we acknowledge the importance and difficulty of obtaining truly private datasets for research purposes. Public datasets are widely used due to accessibility and benchmarking needs, but this introduces challenges in controlling for pre-training overlap. Addressing these limitations remains an important future work [2].
>
>     >[1] Radford, A., Wu, J., Child, R., Luan, D., Amodei, D. and Sutskever, I., 2019. Language models are unsupervised multitask learners. OpenAI blog, 1(8), p.9.
>
>     >[2] Tramèr, F., Kamath, G. and Carlini, N., 2024. Considerations for differentially private learning with large-scale public pretraining. International Conference on Machine Learning
>
>     >[3] Yu, D., Naik, S., Backurs, A., Gopi, S., Inan, H.A., Kamath, G., Kulkarni, J., Lee, Y.T., Manoel, A., Wutschitz, L. and Yekhanin, S., Differentially Private Fine-tuning of Language Models. In International Conference on Learning Representations.
>
>     >[4] Li, X., Tramer, F., Liang, P. and Hashimoto, T., 2023 Large Language Models Can Be Strong Differentially Private Learners. In International Conference on Learning Representations.

---

> > ### Author Response · Authors · 2024-12-20
> > **Response to Reviewer 5EZR's Requested Changes (2/2)**
> >
> > - From Section 5.3 onwards, it should be emphasized in the tables that  δ=10−5
> > >Thank you, we have updated the paper to incorporate this suggestion and marked the revision in blue.
> >
> >
> > - Probably very minor, but I think has a very different meaning from 1e-5; the former is with the mathematical constant , while the latter is .
> > >Thank you, we have updated the paper to incorporate this suggestion and marked the revision in blue.
> > - I also would like to make the following request. If the experimental results are convincing, I will recommend the paper for publication.
> > Could the authors also evaluate the data synthetic methods using pairwise metrics, such as "Pair" and "Corr" in this paper.
> > Could the authors also evaluate previous DP-focused tabular data synthetic methods such as DPSyn, PrivMRF and Private-PGM?
> > >Thank you for your constructive feedback and for suggesting additional experiments to further validate our results. Evaluating the synthetic data using pairwise metrics such as "Pair" and "Corr" is a valuable suggestion, as these metrics can provide insights into the ability of synthetic methods to preserve relationships between features. We will incorporate these evaluations into our analysis and report the outcomes accordingly. Additionally, we will evaluate the feasibility of the suggested DP baselines and report back as soon as we have results.

---

### Review · Reviewer_XUE8 · 2024-12-30

**Summary Of Contributions:**

The paper makes the observation that standard DP tabular fine-tuning suffers from unnecessary private learning of non-private tabular elements, such as table structure, which does not need privacy. Hence, it proposes a two-stage fine-tuning framework where the first stage involves fine-tuning a pre-trained LLM on a pseudo dataset, to learn non-private tabular elements such as tabular structure, then privately fine-tune on the private dataset in the second stage. The paper proposes two methods to generate the pseudo dataset, one using a uniform distribution based on statistics from the private dataset, and another using out-of-distribution public data. The paper demonstrates that the proposed method can outperform all baselines for certain cases. Moreover, the paper experimentally demonstrates that column shuffling negatively affects DP tabular training.

**Audience:**

Yes

**Broader Impact Concerns:**

The current Borader Impact Statement is sufficiently addressed.

**Claims And Evidence:**

Yes

**Requested Changes:**

1. I would like to see a discussion on how the authors handled potential privacy leakage of the uniform pseudo data generation technique in Stage 1 (weakness 1).
2. Address weakness 5.
3. Since the authors claim inference latency improvement in the contributions, I recommend adding more discussion about inference/sampling in the main paper, since I feel like that aspect is lacking and should be emphasized more since it is a strong contribution. In particular, if space permits, move some information about GPT-2 sampling in the Appendix to the background section, to better understand how tabular generation works and where the inference latency could come from. Then move the sampling cost table (Table 7) to the experiments section (Section 5) and add a discussion about why DP-2Stage-U demonstrates significantly faster inference over DP-Standard.
4. Since I couldn’t find a consistent trend for the proposed methods across both datasets (weakness 2)-- for example, DP-2Stage-U performs the worst for F1 compared to DP-2Stage-O and DP-Standard on Adult but performs the best on Airline– it would be great if the authors could include one more dataset to better understand the trends. I understand that it might be computationally expensive to run all baselines, but the inclusion of an additional dataset would be insightful.

**Strengths And Weaknesses:**

Strengths:
1. The paper is well written and the proposed method is understandable.
2. The problem tackled is important, as DP tabular generation is understudied
3. I like this idea/paradigm of first fine-tuning on a pseudo dataset (to help the model learn table structure), then privately fine-tuning on the original dataset. It makes a lot of sense.
4. The code will be open-sourced.
5. The experimental evaluations seem comprehensive; the proposed method was compared against many baselines in the evaluations.
6. The experimental results on the impact of column shuffling for non DP and DP training is interesting and a useful result moving forward with DP training.

Weaknesses:
1. I’m a bit concerned about the uniform pseudo data method, particularly how the range of the numerical columns are selected. Because if the range is calculated based on the private data, then this would leak privacy. For example, if you know the age column of your dataset is bounded between 0-100, and generate pseudo data is using this bound, then the pseudo data contains private information about the original dataset. Hence, the LLM is non-privately fine-tuned on this uniform pseudo data, which would leak privacy. How did the authors control for this?
2. The experimental results in section 5.3 are a bit inconclusive to me. The conclusion I drew is that the proposed method using GPT2 outperforms all baselines for the Airlines dataset, but is outperformed by DP-CTGAN on the Adult dataset (33.4 vs 46.5 F1 for $\epsilon=8$). Furthermore, for no privacy, GPT-2 outperforms CTGAN on Adult (68.9 vs 59.6 F1, respectively). And the proposed method only improves over standard DP by 2 F1 for $\epsilon=8$. Hence, although the proposed method does provide some improvement over standard DP, the gains feel marginal. And the relatively large gap between the proposed method and non-private baseline highlights the need for even more improvement. Also, it is a bit strange that for either dataset, one of the proposed methods performs worse than DP-Standard.
3. Since stage 2 places a higher weight on the value tokens, I’m wondering if one could avoid template learning (skip stage 1) by just optimizing/learning only the token values of the serialized text/table. For example, if you have a serialized record “age is 32”, rather than DP fine-tuning on the entire sequence, which standard DP fine-tuning fails at, could one DP optimize on only the token values of the key-value pairs. I.e., calculate the loss only on “32” using “age is “ as the prefix/prior tokens. Then, during tabular generation, you can use the keys and non-functional elements as the prefix for prompts, e.g., “age is”, then have the LLM generate the value, e.g. “32”. We’d have to assume the column names are public, which is already the case in the uniform pseudo data.
4. The authors briefly remark that DP-2Stage-U achieves the fastest inference time in section 5.4. However, I felt like there wasn’t much discussion/background on the inference aspect of the work (although, I guess, some details are included in the Appendix). Still, it is unclear to me why the DP-2Stage-U framework provides faster inference.
5. It would be great to have a brief discussion/analysis as to why column shuffling does not work for DP training. The authors mention due to gradient perturbations, but a more detailed analysis would make this point more clear.

---

> ### Author Response · Authors · 2025-01-07
> **Response to Reviewer XUEB's Weaknesses (1/2)**
>
> ### Weaknesses
>
> 1. I’m a bit concerned about the uniform pseudo data method, particularly how the range of the numerical columns are selected. Because if the range is calculated based on the private data, then this would leak privacy. For example, if you know the age column of your dataset is bounded between 0-100, and generate pseudo data is using this bound, then the pseudo data contains private information about the original dataset. Hence, the LLM is non-privately fine-tuned on this uniform pseudo data, which would leak privacy. How did the authors control for this?
> > We thank the reviewer for this insightful question which was also raised by Reviewer 5EZR. We proposed DP-2Stage-U as a feasible solution in cases where the range of numerical features as well as categorical features are not sensitive information and can be public. However, in the opposite case as also highlighted in [1], we completely agree that such information must also be derived using differential privacy to ensure that the entire solution is differentially private.  Another alternative we proposed, in cases where this information is indeed considered sensitive, is to use a public dataset, even one of a completely different nature (out-of-distribution), to ensure that the solution remains differentially private. We hope our response addressed your concern, and we would be more than happy to provide further clarification if necessary.
>
>     > [1] Stadler, T., Oprisanu, B. and Troncoso, C., 2022. Synthetic data–anonymisation groundhog day. In 31st USENIX Security Symposium (USENIX Security 22).
>
> 2. The experimental results in section 5.3 are a bit inconclusive to me. The conclusion I drew is that the proposed method using GPT2 outperforms all baselines for the Airlines dataset, but is outperformed by DP-CTGAN on the Adult dataset (33.4 vs 46.5 F1 for ϵ=8). Furthermore, for no privacy, GPT-2 outperforms CTGAN on Adult (68.9 vs 59.6 F1, respectively). And the proposed method only improves over standard DP by 2 F1 for ϵ=8. Hence, although the proposed method does provide some improvement over standard DP, the gains feel marginal. And the relatively large gap between the proposed method and non-private baseline highlights the need for even more improvement. Also, it is a bit strange that for either dataset, one of the proposed methods performs worse than DP-Standard.
> > We thank the reviewer for their question.
> Regarding the observed DP-CTGAN’s inconsistent performance across the two datasets, we would like to note that while DP-CTGAN outperforms GPT-2 models on F1, AUC, and ACC for the Adult dataset, GPT-2 methods demonstrate superior performance on other metrics. To better support our results and incorporate the reviewer’s, and Reviewer 5EZR's suggestions, we have included additional dataset, metrics, and DP-baselines into our analysis (see Requested Changes 4 response).
>
>     >Furthermore, as noted by the reviewer, the gap observed in the non-private experiments highlights the potential for improvement in this area. Our work aims to explore ways to narrow this gap, but more research is required to advance private deep-learning-based synthetic tabular data generators. This remains an important direction for future exploration.
>
> 3. Since stage 2 places a higher weight on the value tokens, I’m wondering if one could avoid template learning (skip stage 1) by just optimizing/learning only the token values of the serialized text/table. For example, if you have a serialized record “age is 32”, rather than DP fine-tuning on the entire sequence, which standard DP fine-tuning fails at, could one DP optimize on only the token values of the key-value pairs. I.e., calculate the loss only on “32” using “age is “ as the prefix/prior tokens. Then, during tabular generation, you can use the keys and non-functional elements as the prefix for prompts, e.g., “age is”, then have the LLM generate the value, e.g. “32”. We’d have to assume the column names are public, which is already the case in the uniform pseudo data.
>  > We thank reviewer XUEB for their very interesting suggestion. Indeed, this approach could be exploited as an alternative to the DP-Standard approach, provided that the keys are public, as the reviewer mentioned. However, we would like to emphasize that the loss associated with the keys is important for optimizing the model and preserving the correlation between the keys and their respective values. Otherwise, the model would be capable of generating different values for each distribution, but without necessarily respecting the corresponding key. In such cases, post-processing would likely be required to map the values back to the correct keys. Exploring this approach in greater detail could provide valuable insights and will be an interesting direction for future work.

---

> > ### Author Response · Authors · 2025-01-07
> > **Response to Reviewer XUEB's Weaknesses (2/2)**
> >
> > 4. The authors briefly remark that DP-2Stage-U achieves the fastest inference time in section 5.4. However, I felt like there wasn’t much discussion/background on the inference aspect of the work (although, I guess, some details are included in the Appendix). Still, it is unclear to me why the DP-2Stage-U framework provides faster inference.
> > > In DP-2Stage-U, the learned format from Stage 1 facilitates faster inference during Stage 2 sampling because, during Stage 1 fine-tuning, the model is trained on task-relevant patterns with keys and values drawn from the same distribution as the private data. This is in contrast to DP-Standard, where shuffling and learning under a noisy signal make the task more complex, often resulting in suboptimal models. For DP-2Stage-O, while shuffling is enabled, the task becomes challenging because the out-of-distribution key/value pairs learned in Stage 1 must be unlearned during Stage 2 DP fine-tuning. Shuffling introduces additional variation, causing some keys and values from Stage 1 to appear in Stage 2 samples. This necessitates multiple sampling iterations to discard these out-of-distribution keys using the imputation sampling method described in the paper.
> > It is also worth noting that when shuffling is disabled, the inference times for all three approaches—DP-2Stage-U, DP-2Stage-O, and DP-Standard—are comparable.
> >
> > 5. It would be great to have a brief discussion/analysis as to why column shuffling does not work for DP training. The authors mention due to gradient perturbations, but a more detailed analysis would make this point more clear.
> > >Our hypothesis is that fixing the shuffling simplifies the task for the model. DP optimization often grants a limited number of training rounds. When the positions of the keys and irrelevant tokens remain consistent, the model can focus solely on learning the values, which is particularly useful under the noisy gradient scenario. As a result, the model's learning process becomes more stable and efficient.
> >
> >     >In contrast, column shuffling increases variability in the data representation. This variability can exacerbate the effects of gradient perturbations, making it harder for the model to converge effectively on meaningful patterns. The noise introduced by DP mechanisms, combined with the added variability from shuffling, likely disrupts the model's ability to generalize well.
> >
> >    >A deeper formal analysis of this phenomenon—especially the interplay between column shuffling, gradient perturbations, and the model's ability to learn under DP constraints—is non-trivial and presents an interesting direction for future research.

---

> > > ### Author Response · Authors · 2025-01-07
> > > **Response to Reviewer XUEB's Requested Changes**
> > >
> > > 1 & 2, please refer to weakness 1& 5 response.
> > >
> > > 3. Since the authors claim inference latency improvement in the contributions, I recommend adding more discussion about inference/sampling in the main paper, since I feel like that aspect is lacking and should be emphasized more since it is a strong contribution. In particular, if space permits, move some information about GPT-2 sampling in the Appendix to the background section, to better understand how tabular generation works and where the inference latency could come from. Then move the sampling cost table (Table 7) to the experiments section (Section 5) and add a discussion about why DP-2Stage-U demonstrates significantly faster inference over DP-Standard.
> > > > We thank the reviewer for this valuable suggestion. While we acknowledge the importance of inference latency as a strong contribution of our work, we believe a more detailed analysis is needed to fully unpack the differences in inference performance when shuffling is enabled versus disabled. For this reason, we decided to keep these discussions in the Appendix to avoid speculative conclusions in the main paper. However, we hope our findings inspire future research to explore this direction more thoroughly.
> > >
> > >     >We will consider moving Table 7 to the experiments section and expanding the discussion about the significant inference speed improvements observed with DP-2Stage-U over DP-Standard and DP-2Stage-O when shuffling is enabled.
> > >
> > >
> > > 4. Since I couldn’t find a consistent trend for the proposed methods across both datasets (weakness 2)-- for example, DP-2Stage-U performs the worst for F1 compared to DP-2Stage-O and DP-Standard on Adult but performs the best on Airline– it would be great if the authors could include one more dataset to better understand the trends. I understand that it might be computationally expensive to run all baselines, but the inclusion of an additional dataset would be insightful.
> > > > We included an additional dataset, Texas [1], following the preprocessing steps described in [2, 3]. The results for this dataset are presented in Table 3 of [this document](https://anonymous.4open.science/r/DP-2Stage-B8B1/docs/reviewer2_req.pdf). New changes are highlighted in blue, and we have also included a description of the dataset in the corresponding section.
> > >
> > >     >Furthermore, we incorporated results for two additional metrics suggested by Reviewer 5EZR: Pair and CorAcc, based on the implementation details provided in [4]. To enhance the evaluation, we included two additional baselines for marginal-based models, AIM [5] and MST [6], as these models have demonstrated superior performance [4].
> > >
> > >    >Since our DP-2Stage-O approach utilizes an out-distribution pseudo-dataset, we have clarified the distinction between pseudo-datasets in Table 3. Specifically, we present results for scenarios where each dataset is treated as a public dataset for the others.
> > >
> > >    > In Table 3, we observe that GPT-2 baselines exhibit better performance on the new dataset. Regardless of the dataset used in Stage 1 for DP-2Stage-O, the results remain competitive compared to Standard-DP across all three datasets. However, DP-2Stage-U shows lower performance on utility metrics (F1, AUC, and ACC) for Texas and Adult datasets compared to DP-2Stage-O and Standard-DP but demonstrates more competitive performance on Pair, CorAcc, and Hist metrics.
> > >
> > >     >Additionally, marginal-based models generally achieve the best performance across most metrics. However, they sometimes perform poorly on Hist and Pair metrics, particularly for the Airline dataset. This is likely due to the higher number of numerical columns in this dataset (refer to Table 1 for dataset statistics and class ratios).
> > >
> > >    >Finally, Table 2 presents the non-DP results for all three datasets, where GPT-2 continues to deliver competitive performance.
> > >
> > >     >[1] https://www.dshs.texas.gov/thcic/
> > >
> > >     >[2] Stadler, T., Oprisanu, B. and Troncoso, C., 2022. Synthetic data–anonymisation groundhog day. In 31st USENIX Security Symposium (USENIX Security 22) (pp. 1451-1468).
> > >
> > >     >[3] Afonja, T., Chen, D. and Fritz, M., 2023, September. MargCTGAN: A “Marginally” Better CTGAN for the Low Sample Regime. In DAGM German Conference on Pattern Recognition (pp. 524-537). Cham: Springer Nature Switzerland.
> > >
> > >     >[4] Tao, Y., McKenna, R., Hay, M., Machanavajjhala, A. and Miklau, G., 2022. Benchmarking differentially private synthetic data generation algorithms. AAAI Workshop on Privacy-Preserving Artificial Intelligence.
> > >
> > >     >[5] McKenna, R., Mullins, B., Sheldon, D. and Miklau, G., 2022. AIM: an adaptive and iterative mechanism for differentially private synthetic data. Proceedings of the VLDB Endowment.
> > >
> > >     >[6] McKenna, R., Miklau, G. and Sheldon, D., 2021. Winning the NIST Contest: A scalable and general approach to differentially private synthetic data. Journal of Privacy and Confidentiality.

---

> > > > ### Comment · Reviewer_XUE8 · 2025-01-13
> > > > **Response to Authors**
> > > >
> > > > I thank the authors for their thorough response! It's still not completely clear to me what the exact performance trends look like for the proposed methods, since they could significantly vary on different datasets (e.g. DP-2Stage-U shows lower performance on utility metrics (F1, AUC, and ACC) for Texas and Adult datasets), and there are quite a bit of metrics that are introduced now (both the original utility metrics and the newly introduced ones, CorAcc and Pair). Breaking up Table 3 somehow could help make the results more easily understandable.
> > > >
> > > > However, the authors have addressed my questions/concerns. I have no further questions or concerns to ask at this time. I believe this paper has interesting contributions and results.

---

> > > > > ### Author Response · Authors · 2025-01-23
> > > > > **Thank you**
> > > > >
> > > > > We sincerely thank the reviewer for their thoughtful feedback and for recognizing the contributions and results of our work. We appreciate your suggestion regarding improving our presentation of Table 3 of the new results and will incorporate this feedback into our revised edition. Thank you once again for your valuable input, which has greatly contributed to enhancing the quality and clarity of our work.

---

### Review · Reviewer_6jAQ · 2025-01-07

**Summary Of Contributions:**

This paper develops a two-stage procedure for training differentially private tabular data generators. The general process is training a model based on a non-private pseudo dataset. Then the DP fine-tuning is implemented on a private dataset.  Extensive empirical results are presented for showing the superiority of this framework. Overall, I think this paper is well presented.

**Audience:**

Yes

**Broader Impact Concerns:**

At this stage, we believe that the work presented in this paper does not pose significant ethical or societal concerns.

**Claims And Evidence:**

Yes

**Requested Changes:**

(1) Figure 3 presents various aspects of perplexity. However, this can be confusing for readers unfamiliar with the metric. I suggest that the authors include the formulas for perplexity in the main text for better clarification.

(2) When evaluating the quality of synthetic data, the authors focus on assessing the utility of generative data in downstream learning tasks. Another commonly used approach is to evaluate the fidelity of generative data except statistical divergence metrics. Such as classification method for discriminating real and synthetic data. I suggest that the authors include this type of fidelity evaluation metric in the paper.

(3) When utilizing private statistics from the private dataset for generating a pseudo dataset, the privacy budget for this part is not considered.

(4) The authors should add experiments about the case that the pseudo dataset is significantly different from the private one.

**Strengths And Weaknesses:**

Strength 1: The experiments considered in this paper are thorough, showing significant improvements compared to standard DP training with DPSGD.

Strength 2: DP fine-tuning of LLMs is an exciting and promising research area. This paper makes a valuable contribution by exploring DP tabular generation in this innovative way.

Weakness 1: A key challenge in the problem is that the non-private dataset may not resemble the real dataset. Additionally, in practice, it would be difficult to find a public dataset that has a similar data structure to the private dataset.

Weakness 2: If the public dataset deviates significantly from the private dataset, the advantage over standard DPSGD may be weakened. I think certain experiments should be added to address this aspect.

---

> ### Author Response · Authors · 2025-01-07
> **Response to Reviewer 6jAQ's Weaknesses**
>
> ### Weaknesses
> 1. A key challenge in the problem is that the non-private dataset may not resemble the real dataset. Additionally, in practice, it would be difficult to find a public dataset that has a similar data structure to the private dataset.
> > We appreciate the reviewer's observation. We would like to highlight that our proposed DP-2Stage-O method is specifically designed to address this scenario, where the data used in Stage 1 originates from an out-of-distribution source. As shown in Table 3 of [the document](https://anonymous.4open.science/r/DP-2Stage-B8B1/docs/reviewer2_req.pdf), we have provided results using various out-of-distribution datasets for DP-2Stage-O, demonstrating that it consistently outperforms DP-Standard.
>
>
> 2. If the public dataset deviates significantly from the private dataset, the advantage over standard DPSGD may be weakened. I think certain experiments should be added to address this aspect.
> > Please see response above.

---

> > ### Author Response · Authors · 2025-01-08
> > **Response to Reviewer 6jAQ's Requested Changes**
> >
> > 1. Figure 3 presents various aspects of perplexity. However, this can be confusing for readers unfamiliar with the metric. I suggest that the authors include the formulas for perplexity in the main text for better clarification.
> > > We thank the reviewer for the helpful suggestion. The definition and formula for perplexity are currently provided in Appendix C. Could the reviewer kindly confirm if they noticed this? If the reviewer believes it would be more beneficial to include this information in the main text, we are happy to consider moving it accordingly.
> >
> > 2. When evaluating the quality of synthetic data, the authors focus on assessing the utility of generative data in downstream learning tasks. Another commonly used approach is to evaluate the fidelity of generative data except statistical divergence metrics. Such as classification method for discriminating real and synthetic data. I suggest that the authors include this type of fidelity evaluation metric in the paper.
> > > We appreciate the reviewer’s suggestion, which aligns closely with the feedback provided by Reviewer 5EZR. In response, we have implemented two additional metrics, CorAcc and Pair, to better evaluate the quality of the synthetic datasets (please see [this pdf](https://anonymous.4open.science/r/DP-2Stage-B8B1/docs/reviewer2_req.pdf)). Could the reviewer kindly confirm if these additions address their concern satisfactorily?
> >
> > 3. When utilizing private statistics from the private dataset for generating a pseudo dataset, the privacy budget for this part is not considered.
> > > We thank the reviewer for this insightful question which was also raised by Reviewer 5EZR and XUEB. We proposed DP-2Stage-U as a feasible solution in cases where the range of numerical features as well as categorical features are not sensitive information and can be public. However, in the opposite case as also highlighted in [1], we completely agree that such information must also be derived using differential privacy to ensure that the entire solution is differentially private.  Another alternative we proposed, in cases where this information is indeed considered sensitive, is to use a public dataset, even one of a completely different nature (out-of-distribution), to ensure that the solution remains differentially private. We hope our response addressed your concern, and we would be more than happy to provide further clarification if necessary.
> >
> >     > [1] Stadler, T., Oprisanu, B. and Troncoso, C., 2022. Synthetic data–anonymisation groundhog day. In 31st USENIX Security Symposium (USENIX Security 22).
> >
> >
> > 4. The authors should add experiments about the case that the pseudo dataset is significantly different from the private one.
> > >We thank the reviewer for this suggestion. Please refer to our response to Weakness 1. We believe the experiments and analysis provided for DP-2Stage-O cover this case effectively.

---

### Review · Reviewer_hmKd · 2025-01-08

**Summary Of Contributions:**

This paper proposes a method for tabular data generation using LLM fine-tuning, particularly in combination with differential privacy (DP), and explores its scalability through experiments.

**Audience:**

Yes

**Broader Impact Concerns:**

None.

**Claims And Evidence:**

No

**Requested Changes:**

Please see above.

**Strengths And Weaknesses:**

This paper introduces a method for generating private tabular data by fine-tuning large language models (LLMs). It is likely to attract attention within the LLM and Differential Privacy (DP) communities. However, the work has several notable weaknesses that need to be addressed:
1. The paper introduces two versions of DP-2Stage, but it is unclear which version is the primary proposal. This ambiguity makes it difficult for readers to discern the main contributions and focus of the study.
2. The Abstract and Introduction require more technical depth. For example, the abstract should elaborate on specific challenges encountered and performance results across various settings. Similarly, the introduction should discuss scalability concerns and other practical aspects of the approach.
3. The paper posits some intuitive hypotheses but fails to substantiate them with sufficient experimental evidence. For instance, the authors suggest that inefficient allocation of privacy budgets in tabular data structures is the key reason for poor performance. However, this hypothesis is not empirically verified. Extending experiments to more general data generation settings, beyond tabular data, could significantly strengthen the paper's concreteness and contributions.
4. The authors mentioned that "this work does not analyze the impact of pseudo-data used in the first stage and only examines a different family of LLMs and fine-tuning strategies". However, the addition of a new LLM type and fine-tuning method represents only a minor improvement and lacks originality. Furthermore, the proposed technique performs worse than established baselines like DP-CGAN and DP-GAN, as shown in Table 3. While the authors claim improvements in learning cost over these baselines, this argument could have been more compelling if the introduction had framed the study’s goals around optimizing Privacy, Accuracy, and Cost simultaneously.
5. Critical analyses are missing in some sections. For example:
- In Figure 3, the reason why the blue line performs better than the non-DP counterpart is not explained.
- For the HIST metric, the trend differs from other metrics in Tables 2 and 4, but no rationale is provided for this discrepancy.
6. To improve the reader's understanding, the Introduction section should include the storyline below:
1) lack of tabular data. - 2) privacy concern when exchanging directly - 3) overcome via privacy-preserving data generator
7. Minor error corrections:
- In Figure 3, some subfigures are missing the blue curve.
- References are missing in the first sentence of the Introduction section.
- Function f should include the variable c in Definition 1.
- An explanation is needed for why the curves converge with respect to steps in Figure 3.
- I recommend the authors to add additional metrics for membership inference attacks, such as the attack success rate.
- Bullets 1 and 2 of the contributions are incomplete sentences.

---

> ### Author Response · Authors · 2025-01-23
> **Response to Reviewer hmKd's Weaknesses (1/2)**
>
> 1. The paper introduces two versions of DP-2Stage, but it is unclear which version is the primary proposal. This ambiguity makes it difficult for readers to discern the main contributions and focus of the study.
> > Thank you for your feedback. Our intention is not to propose one version of DP-2Stage as the primary solution but to present the two-stage fine-tuning process as a flexible and effective approach for LLM-based private synthetic tabular data generation. The results show that both versions (DP-2Stage-O and DP-2Stage-U) perform well across most datasets and metrics, demonstrating the overall value of our approach in addressing this problem. We hope this clarifies the focus of our study.
>
> 2. The Abstract and Introduction require more technical depth. For example, the abstract should elaborate on specific challenges encountered and performance results across various settings. Similarly, the introduction should discuss scalability concerns and other practical aspects of the approach.
> > Thank you for your thoughtful feedback. We appreciate your suggestions regarding the abstract and introduction. While we aimed to provide sufficient information to motivate the problem and situate our work within the broader context, we understand the value of incorporating additional technical depth. We will carefully revisit these sections to ensure all relevant aspects, including the challenges, performance results, and practical considerations, are adequately addressed.
>
> 3. The authors mentioned that "this work does not analyze the impact of pseudo-data used in the first stage and only examines a different family of LLMs and fine-tuning strategies". However, the addition of a new LLM type and fine-tuning method represents only a minor improvement and lacks originality. Furthermore, the proposed technique performs worse than established baselines like DP-CGAN and DP-GAN, as shown in Table 3. While the authors claim improvements in learning cost over these baselines, this argument could have been more compelling if the introduction had framed the study’s goals around optimizing Privacy, Accuracy, and Cost simultaneously.
> > We thank the reviewer for their feedback and comments. We would like to emphasize that the results we presented, including those on additional metrics and datasets, demonstrate that our approach outperforms established baselines such as DP-CTGAN and DP-GAN across the majority of metrics and datasets. For further details, we kindly refer the reviewer to [this document](https://anonymous.4open.science/r/DP-2Stage-B8B1/docs/reviewer2_req.pdf). If there are any specific concerns or questions regarding these results, we are happy to provide further clarifications.
> Regarding the framing of our study’s goals, we appreciate the reviewer’s perspective on the potential for broader exploration. However, we intentionally chose to focus on privacy optimization in this work, given the design and scope of our current experiments. We acknowledge that a more comprehensive exploration of Privacy, Accuracy, and Cost could be valuable should be considered in future studies.

---

> > ### Author Response · Authors · 2025-01-23
> > **Response to Reviewer hmKd's Weaknesses (2/2)**
> >
> > 4. Critical analyses are missing in some sections. For example: In Figure 3, the reason why the blue line performs better than the non-DP counterpart is not explained.
> >
> > > We appreciate the thoughtful feedback. We discuss the point in detail below and will include the discussion in our manuscript.
> >
> > > The reduced perplexity of the blue line (DP-2Stage-U) in Figure 3, particularly on keys and other tokens, is comparable to that of the non-DP model due to the use of pseudo-data in stage 1. To elaborate, DP-2Stage-U leverages table features, categories, and statistics derived from private data to train the first stage model. When fine-tuned with shuffling disabled, the model efficiently learns keys and other tokens due to the reduced variation, leading to a rapid decrease in perplexity. During the second stage, despite training with DP, the model retains the low perplexity of task-relevant patterns learned in the first stage, achieving performance comparable to the non-DP counterpart on the keys and other tokens.
> > In contrast, DP-2Stage-O uses a different dataset during the first stage, exposing the model to new tokens in the second stage that must be learned. Nevertheless, the learned patterns during the first stage contribute to reduced perplexity on key and value tokens compared to DP-Standard. This difference highlights the strengths of the proposed two-stage approach across different configurations.
> >
> > 5. For the HIST metric, the trend differs from other metrics in Tables 2 and 4, but no rationale is provided for this discrepancy.
> > > Table 2 presents results for the non-DP setting, which is inherently less challenging than the DP setting represented in Table 4. The HIST metric, like other metrics, reflects the impact of training with differential privacy, which introduces additional constraints and variability compared to the non-DP scenario.
> >
> > 6. To improve the reader's understanding, the Introduction section should include the storyline below:
> > lack of tabular data. - 2) privacy concern when exchanging directly - 3) overcome via privacy-preserving data generator
> > > The introduction is currently structured as (1) The importance of tabular data, (2) Privacy concerns related to sharing tabular data, (3) The use of private synthetic data to address these challenges, (4) Limitations of existing methods, (5) The potential of pre-trained LLMs to revolutionize various domains, including tabular data generation, (6) There's limited work in leveraging pre-trained LLMs for tabular data generation with DP, (7) Challenges posed by directly fine-tuning with DP due to inefficient privacy budget allocation when learning tabular data structures, and (8) Our solution: a two-stage learning approach designed to progressively capture tabular data structures.
> >
> > Thank you for taking the time to review our work and provide feedback. Please let us know if we have sufficiently addressed your questions, or if there are any further points you would like us to clarify.

---

### Decision · Action_Editor_Tbsd · 2025-02-02

**Recommendation:** Accept with minor revision

**Comment:**

The paper introduces a two-stage fine-tuning approach for differentially private (DP) tabular data generation using large language models (LLMs). It first fine-tunes on a pseudo dataset to learn tabular structures, then applies DP fine-tuning on private data for privacy protection. This method improves performance over existing DP approaches.


The paper would be of interest to researchers working on (synthetic) tabular data generation. According to the reviewers, the paper is well-structured, and provides empirical results to support its claims. While some concerns existed regarding experimental consistency, the authors have addressed reviewer concerns. Minor improvements, such as refining result interpretations, adding clarifications on privacy guarantees (in Stage 1), and structuring results more clearly, would enhance the paper. Please consider the final minor comments of the reviewers (e.g. clarification/presentation suggestions from reviewer's XUEB).

**Audience:**

Yes, the paper would be of interest to researchers working on (synthetic) tabular data generation (which appears in different applications).

**Claims And Evidence:**

Yes, the claims are generally supported by evidence. There were some initial concerns about the inconsistencies and performance across different datasets, which was resolved during during the discussion.

---

> ### Author Response · Authors · 2025-03-03
> **Camera ready**
>
> We sincerely appreciate our TMLR reviewers for their insightful discussions and suggestions, which have undoubtedly improved the quality of our work. We also extend our deep gratitude to the Action Editor for their support, dedication, and for recommending our work for acceptance. Additionally, we are thankful for the wonderful TMLR community, which fosters a smooth, productive, and respectful review process.
>
> We have uploaded the camera-ready version incorporating the suggestions such as bringing up the additional dataset and metrics into the main paper body, refining result interpretation and structuring result more clearly. We have also expanded our discussion on privacy guarantees in Stage 1 and incorporated various feedback, such as including delta values when presenting DP results, and adding the DP-Standard definition in Appendix A.
>
> We hope these changes meet your expectations and look forward to your feedback.

---

> > ### Comment · Action_Editor_Tbsd · 2025-03-03
> > **Syle**
> >
> > It seems the submitted file does not follow the TMLR stylefile exactly. There should be link to the OpenReview page and also a heading. Please correct that.

---

> > > ### Author Response · Authors · 2025-03-03
> > > **Style corrected**
> > >
> > > Thank you,